# POINT-MOE: LARGE-SCALE MULTI-DATASET TRAINING WITH MIXTURE-OF-EXPERTS FOR 3D SEMANTIC SEGMENTATION

**Xuweiyi Chen**[1]     **Wentao Zhou**[1]     **Aruni RoyChowdhury**[2]     **Zezhou Cheng**[1]

[1]University of Virginia     [2]MathWorks

https://point-moe.cs.virginia.edu/

## ABSTRACT

While massively scaling both data and models have become central in NLP and 2D vision, their benefits for 3D point cloud understanding remain limited. We study the initial step of scaling 3D point cloud understanding under a realistic regime: large-scale multi-dataset joint training for 3D semantic segmentation, with no dataset labels available at training or inference time. Point clouds arise from a wide range of sensors (*e.g.*, depth cameras, LiDAR) and scenes (*e.g.*, indoor, outdoor), yielding heterogeneous scanning patterns, sampling densities, and semantic biases; naively mixing such datasets degrades standard models. Therefore, we introduce **Point-MoE**, a Mixture-of-Experts design that expands model capacity through sparsely activated expert MLPs and a lightweight top-$k$ router, allowing tokens to select specialized experts without requiring dataset supervision. Trained jointly on a diverse mix of indoor and outdoor datasets, and evaluated on seen datasets as well as in zero-shot settings, Point-MoE outperforms prior methods without using dataset labels for either training or inference. This outlines a scalable path for 3D perception: letting the model discover structure in heterogeneous 3D data rather than imposing it via manual curation or dataset-specific heuristics.

## 1 INTRODUCTION

In both language and vision, we have witnessed remarkable advances driven by two trends: the aggregation of massive, heterogeneous datasets [4, 8, 58, 61, 66, 97], and the deployment of ever-larger models trained on them in a unified manner [7, 16, 21, 36, 51, 57]. These models succeed not because they are crafted for a specific dataset, but because they learn underlying regularities across multiple datasets. However, 3D point cloud understanding has not yet followed the same trajectory, despite its wide applications in robotics [11, 53, 90], autonomous systems [12, 13, 43], augmented reality [6, 77], and embodied intelligence [72, 79]. Taking 3D semantic segmentation as an example, although diverse 3D datasets exist—such as ScanNet [19], SemanticKITTI [5], nuScenes [9], and ASE [2]—each covers only a narrow slice of real-world variation. Point clouds arise from diverse pipelines (RGB-D, LiDAR, multi-view stereo), each with their own distinct densities, artifacts, and semantic biases (Fig. 1, left). The result is a rich yet siloed ecosystem of 3D models, where models trained on one domain often fail to generalize to others.

A natural response is to *train across silos*, presumably resulting in a single set of model parameters that is effective across the spectrum of datasets. One might, for example, naively train the state-of-the-art Point Transformer V3 (PTv3) [81] (Fig. 1a) on a mixture of datasets from multiple domains, *e.g.* indoor and outdoor. In practice, however, such models struggle to reconcile the heterogeneity of the data distribution even as capacity scales (Tab. 1). To mitigate this, recent methods have introduced dataset-aware components. Point Prompt Training (PPT) [82] (Fig. 1b) assigns dataset-specific normalization layers, while One-for-All [75] uses a lightweight dataset classifier with dataset-specific adapters. While effective, these methods rely on dataset labels, identifying which specific dataset an input sample originated from, during training and inference. Moreover, adapting only the normalization parameters may be insufficient for large-scale multi-dataset training.

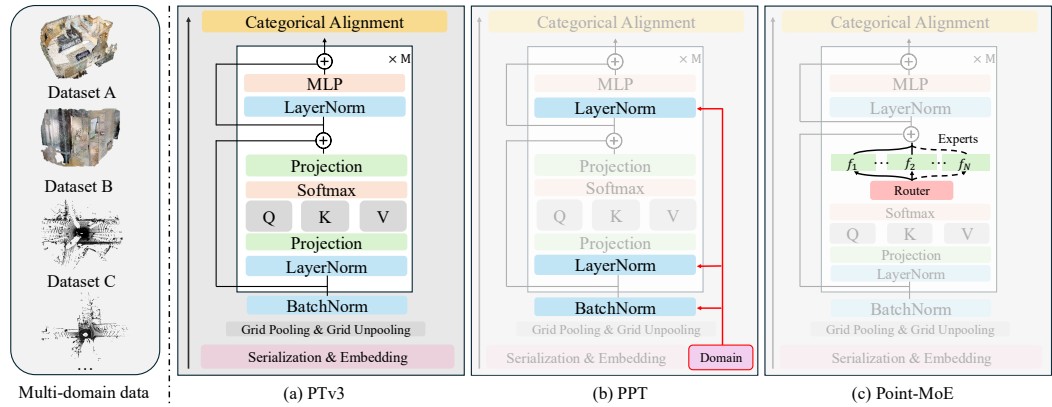

Figure 1: **Overview of multi-datasets training architectures.** Point clouds exhibit diverse characteristics across datasets. **(a)** Naively training Point Transformer V3 (PTv3) [81] on multi-datasets data leads to degraded performance within each domain. **(b)** Point Prompt Training (PPT) [82] addresses this by adding dataset-aware normalization parameters. **(c)** Our proposed Point-MoE tackles this challenge with Mixture-of-Experts (MoE), enabling dynamic expert specialization across datasets.

At deployment, point clouds arrive from mixed, unlabeled sources with unknown provenance, *i.e.* there is no oracle "dataset ID" available during inference. Accordingly, we study a realistic regime: **large-scale multi-dataset joint training** for 3D semantic segmentation with **no dataset labels at inference**. We hypothesize that Mixture-of-Experts (MoE) architectures [30, 34] are well-suited to this setting. We introduce **Point-MoE**, a sparse MoE model built upon PTv3 [81]. As illustrated in Fig. 1c, Point-MoE replaces each attention projection layer in each PTv3 block with a MoE module composed of expert MLPs and a lightweight router that selects a sparse top-$k$ subset of experts per token. Our design is motivated by three insights. First, MoE encourages expert specialization and token-to-expert routing [15, 87], enabling a single model to learn from mixed datasets without access to dataset labels at training or test time. Second, compared to normalization-based adaptations such as PPT [82] and One-for-All [75], MoE offers far more expressive capacity to model dataset-specific variations. Third, sparse MoE architectures maintain computational efficiency during training and inference, and are a standard mechanism for scaling large models in NLP and 2D vision [20, 22, 24, 40, 59]. We extend these benefits, well-established in language and 2D vision, to 3D point cloud understanding, focusing on 3D semantic segmentation. We conduct extensive experiments across a diverse collection of 3D datasets spanning indoor and outdoor scenes, synthetic and real datasets, and multiple sensor modalities. We evaluate on both in-domain and zero-shot datasets, all without dataset labels. To comprehensively assess Point-MoE, we evaluate both in-domain (seen) datasets and out-of-domain (zero-shot) datasets, all without access to dataset labels. Our goal is to demonstrate that Point-MoE not only achieves strong performance on seen datasets but also generalizes robustly to unseen distributions, without relying on dataset-specific augmentation or supervision. Across both regimes, Point-MoE consistently outperforms competitive multi-dataset baselines. We further ablate the number of experts, sparsity level, and MoE placement (Tab. 3), and find that Point-MoE learns coherent routing patterns and exhibits organic expert specialization driven by the underlying data distribution (Fig. 2, 3, 4).

To the best of our knowledge, this work provides the first systematic study of MoE for 3D point cloud understanding under the large-scale multi-dataset training regime. Our contributions are: (1) we focus on multi-dataset 3D semantic segmentation without domain labels, using a single model, and put forth Mixture-of-Experts as a natural solution to this task; (2) we carefully explore the MoE design space and provide extensive ablations that highlight the trade-offs and inform effective configurations; (3) we demonstrate that Point-MoE achieves state-of-the-art performance across seven datasets under this protocol while maintaining inference efficiency; and (4) we analyze token-level routing trajectories and expert usage, revealing encoder–decoder specialization patterns consistent with 3D geometry. Together, these findings point toward a scalable path for 3D perception: rather than building separate models for each dataset or domain, a single unified system can adapt across the spectrum of 3D data sources, leveraging scaling laws for data and compute [68].

## 2 RELATED WORK

**Point Cloud understanding.** Deep learning backbones for 3D scene understanding can be broadly categorized by how they process point clouds: projection-based methods [13, 38, 41, 64], voxel-based methods [17, 25, 49, 63], and point-based methods [54, 55, 69, 93]. More recently, transformer-based architectures operating on point clouds have emerged as strong contenders in this space [27, 80, 81, 94]. Among them, Point Transformer V3 (PTv3) [81] represents the current state of the art, introducing a novel space-filling curve based serialization scheme, that transforms the unstructured 3D points into a 1D sequence, enabling efficient transformer-based modeling. Recent works [32, 39, 82, 83, 96] scale point cloud understanding models to multi-dataset settings and show that large-scale point cloud data can improve performance on specific datasets: Sonata [83] explores multi-dataset self-supervised learning, but does not train across indoor and outdoor datasets. On the other hand, Point Prompt Training (PPT) [82] demonstrates that using dataset-specific normalization layers during pretraining improves 3D semantic segmentation performance across multiple datasets along with dataset-specific fine-tuning. In 3D detection, One-for-All [75] adopts a sparse convolutional network with dataset-specific partitions and normalization layers to boost multi-datasets performance, and UniDet3D [37] achieves state-of-the-art performance with multi-dataset joint training by merging label spaces across datasets into unified taxonomy and building upon vanilla transformer [71] architecture. Rather than relying on dataset-specific preprocess or architectural branches [75, 82], we propose a unified sparse expert framework that learns to specialize dynamically across various datasets, without requiring explicit dataset labels during training or inference.

**Mixture-of-Experts.** Mixture-of-Experts (MoE) [30, 34] increases model capacity without proportional compute by activating only a subset of experts per input [62]. This conditional computation has powered trillion-scale NLP models [20, 22, 24, 33, 40, 46, 47, 98] and demonstrated similar efficiency in vision [59]. Beyond scaling, MoE is naturally suited for heterogeneous or multi-datasets data, as experts tend to self-organize by dataset or task. Prior work has shown improvements in multi-source sentiment analysis and POS tagging [26], dataset-adaptive language modeling [28], automatic chart understanding [87], and multilingual translation [15]. Mod-Squad [14] introduced modular expert routing for multi-task learning, balancing cooperation and specialization. Similar trends are observed in vision and vision-language applications: DAMEX [31], MoE-LLaVA [44] and DeepSeek-VL [48, 85]. To our knowledge, this is the first work to extend MoE to multi-dataset training in 3D point clouds. MoE3D [76] recently introduces a MoE formulation for 3D reconstruction.

**Domain generalization.** Classical domain generalization (DG) trains on one or more labeled source domains and targets zero-shot performance on a specified target domain, typically in sensor- and semantic-ontology-homogeneous outdoor LiDAR settings [35, 60, 84, 86, 88, 92]. We instead study large-scale joint training on heterogeneous datasets with naturally misaligned label spaces, evaluating a single model on both seen and zero-shot datasets. Unlike DG, our goal is a single unified model for heterogeneous multi-dataset learning rather than transfer to a fixed target domain, although DG methods remain complementary and could be layered onto Point-MoE as a natural extension.

## 3 POINT-MOE FOR MULTI-DATASET POINT CLOUD TRAINING

### 3.1 LARGE-SCALE MULTI-DATASET JOINT TRAINING

**Task definition.** Given a 3D scene represented as a point cloud $\mathcal{P} = \{p_i\}_{i=1}^{M}$, where each point $p_i \in \mathbb{R}^3$ denotes XYZ coordinates, semantic segmentation aims to predict a single class label $\hat{y}_i \in \mathcal{C}$ for each point $p_i$ from a fixed label set $\mathcal{C} = \{c_1, c_2, \ldots, c_m\}$. Predicted labels for the entire point cloud are compactly denoted as $\hat{\mathbf{y}}$. Similarly, we denote the corresponding ground-truth labels by $\mathbf{y} = [y_1, y_2, \ldots, y_M], y_i \in \mathcal{C}$, where $y_i$ is the true semantic label of point $p_i$.

Let a dataset containing multiple point clouds be denoted as $\mathcal{D} = \{(\mathcal{P}_t, \mathbf{y}_t)\}_{t=1}^{|\mathcal{D}|}$. The complete training set across multiple datasets is then $\mathbb{D} = \{\mathcal{D}_j\}_{j=1}^{|\mathbb{D}|}$. The goal of our multi-dataset joint training is to learn a unified model (*i.e.* a single set of model parameters, without resorting to per-dataset fine-tuning) over all datasets that minimizes prediction error across $\mathbb{D}$, while also generalizing effectively to zero-shot datasets drawn from distributions unseen at training time.

Let $\Phi_\theta$ denote a segmentation model with parameters $\theta$, that takes in a point cloud $\mathcal{P}$ and predicts per-point class labels $\hat{\mathbf{y}}$. Let $\mathcal{L}(\mathbf{y}, \hat{\mathbf{y}})$ denote the loss, *e.g.* cross-entropy, on a point cloud between ground-truth segmentation labels $\mathbf{y}$ and predicted labels $\hat{\mathbf{y}}$. Formally, the training objective is $\min_\theta \ \mathbb{E}_{\mathcal{D}_j \sim \mathbb{D}} \ \mathbb{E}_{(\mathcal{P}, \mathbf{y}) \sim \mathcal{D}_j}[\mathcal{L}(\mathbf{y}, \ \Phi_\theta(\mathcal{P}))]$. We do not assume oracle dataset labels at training or inference. In other words, we do not know which dataset a sample is drawn from.

We can formally contrast this to alternate multi-dataset objective formulations such as [75, 82], that learn an explicit set of dataset-specific parameters $\omega_j$, along with $\theta$: $\min_{\theta, \{\omega_j\}} \ \mathbb{E}_{\mathcal{D}_j \sim \mathbb{D}} \ \mathbb{E}_{(\mathcal{P}, \mathbf{y}) \sim \mathcal{D}_j}\big[\mathcal{L}(\mathbf{y}, \ \Phi_{\theta, \omega_j}(\mathcal{P}))\big]$. PPT [82] instantiates $\{\omega_j\}_{j=1}^{|\mathbb{D}|}$ as the learnable gain and bias terms of the normalization layers of PTv3, trained on a per-dataset basis. During inference, the source dataset $\mathcal{D}_j$ of an incoming point cloud $\mathcal{P}$ must be identified in order to apply the correct normalization $\omega_j$, usually obtained from a lightweight dataset classifier as in One-for-All [75].

**Language-guided classification.** To bridge label discrepancies across datasets (*i.e.* $\mathcal{C}_i \neq \mathcal{C}_j$ for datasets $\mathcal{D}_i$ and $\mathcal{D}_j$), we project per-point features to a shared language space using CLIP [57] text embeddings, following PPT [82]. This enables supervision via class names, without relying on dataset labels. For instance, the class "pillow" is missing in ScanNet (grouped as "other") but explicitly labeled in Structured3D; aligning to the CLIP embedding of "pillow" transfers knowledge across datasets even without direct supervision. All baselines and our Point-MoE are adapted to this language-guided protocol, ensuring fair comparison.

## 3.2 POINT-MOE

**Mixture-of-Experts (MoE) layer.** Mixture-of-Experts [30] is a conditional computation architecture enabling significant capacity expansion while only modestly increasing computational cost. The core of MoE is to route input tokens dynamically to specialized subnetworks, referred to as experts, via a lightweight gating mechanism. Given an input feature vector $\mathbf{x} \in \mathbb{R}^D$, each MoE layer contains $N$ experts $\{f_i\}_{i=1}^N$ where $f_i : \mathbb{R}^D \to \mathbb{R}^D$ and a gating network $G : \mathbb{R}^D \to \mathbb{R}^N$. For each input $\mathbf{x}$, the MoE layer output is computed as a weighted sum over the top-$k$ activated experts:

$$\text{MoE}(\mathbf{x}) = \sum_{i \in \mathcal{S}_\mathbf{x}} G_i(\mathbf{x}) \, f_i(\mathbf{x}), \tag{1}$$

where $\mathcal{S}_\mathbf{x}$ is the top-$k$ expert set and $G_i(\mathbf{x})$ are normalized gate weights. In practice, $G$ is a linear projection followed by a sparse softmax, activating only $k \ll N$ experts. Importantly, MoE introduces a large design space, including the choice of which components to replace, the number of experts, the routing strategy, whether to do auxiliary balancing loss and whether to use top-1 or top-2 gating. We systematically study how different MoE configurations affect performance on 3D point clouds through detailed ablations in the experiments.

**Integration into PTv3.** Given a point cloud $\mathcal{P}$ with $M$ points, PTv3 [81] serializes and embeds the input into a token sequence $\mathbf{Z} = [z_1, \ldots, z_M]$. PTv3 then applies stacked multi-head self-attention (MHSA) blocks interleaved with up- and down-sampling for multi-scale reasoning on these per-token embeddings. At any intermediate layer $\ell$, let $\mathbf{x}_\ell \in \mathbb{R}^{M \times D}$ denote the token sequence at layer $\ell$, where $D$ is the feature dimension, $H$ is the number of heads in multi-head attention, and FFN denotes a feed-forward network. The computations of a self-attention block using standard QKV-formulation and pre-norm residuals is as follows:

$$\mathbf{X} = \text{Norm}(\mathbf{x}_{\ell-1}), \qquad \mathbf{Q} = \mathbf{X}W_q, \ \mathbf{K} = \mathbf{X}W_k, \ \mathbf{V} = \mathbf{X}W_v, \tag{2}$$

$$\mathbf{A} = \text{Softmax}\big(\mathbf{Q}\mathbf{K}^\top / \sqrt{d_h}\big) \mathbf{V}, \quad d_h = D/H, \tag{3}$$

$$\mathbf{O} = W_o \mathbf{A}, \tag{4}$$

$$\mathbf{x}'_\ell = \mathbf{x}_{\ell-1} + \mathbf{O}, \tag{5}$$

$$\mathbf{x}_\ell = \mathbf{x}'_\ell + \text{FFN}\big(\text{Norm}(\mathbf{x}'_\ell)\big) \tag{6}$$

The learnable query, key, and value projections are denoted as $W_q$, $W_k$ and $W_v$, while $W_o$ denotes the projection applied on the attention output $\mathbf{A}$. We apply MoE only at the *attention output projection* ($W_o$) in every block, rewriting Eq. (4) as $\mathbf{O} = \text{MoE}(\mathbf{A})$ ; the query, key, and value projections ($W_q, W_k, W_v$) remain dense. Empirically, placing MoE at $W_o$ yields better performance than placing it in the FFN under the same expansion (Tab. 3d). We hypothesize that the gain comes from adding

nonlinearity and capacity on the projection path: with purely linear projections, the composition of $W_v$ and $W_o$ can be merged into a single projection [47].

Beyond MoE-layer placement, we employ *mixed-dataset training*: each minibatch jointly samples from both indoor and outdoor data (Tab. 4). Having tokens from multiple datasets within a single minibatch, and thus contributing to the gradients of one update step, is useful in facilitating cross-sample interaction and emergent expert specialization.

To illustrate the general applicability of this framework, we further extend MoE to OA-CNNs [52], a sparse convolution backbone. MoE-augmented OA-CNNs likewise outperform the dense version, further underscoring the effectiveness and general applicability of MoE as it is not tied to a specific backbone architecture. Detailed results are in the supplementary material under Sec. B.2.

## 4 EXPERIMENTS

We conduct comprehensive experiments to evaluate Point-MoE for 3D semantic segmentation. First, we describe baseline methods and experimental setups (Sec. 4.1). Then, we present detailed empirical results under two primary joint training scenarios: (1) indoor-only, and (2) combined indoor-outdoor training (Sec. 4.2). Finally, we provide ablation studies that highlight key design choices of the MoE architecture (Sec. 4.3) and detailed analysis of MoE behaviors (Sec. 4.4).

### 4.1 EXPERIMENTAL SETUP

**Datasets.** We conduct experiments on three indoor benchmarks: ScanNet [19], S3DIS [1], and Structured3D [95], and additionally evaluate zero-shot on Matterport3D [10]. Specifically, we train a single model on ScanNet, S3DIS, and Structured3D, and directly evaluate its performance on both seen and unseen datasets via the language-guided head using the same checkpoint. We further test the best-performing Point-MoE architectures and baseline configurations in a more challenging indoor and outdoor joint setting that incorporates nuScenes [9] and SemanticKITTI [5] as training data, and Waymo [67] for zero-shot evaluation. The same evaluation protocol is used across both settings.

**Baselines.** Since Point-MoE is built on PTv3 [81], we adopt PTv3 and PPT [82] as our primary baselines. PPT, originally designed with dataset-specific normalization and per-dataset fine-tuning, is adapted here to the joint training setting. Because PPT requires dataset labels at inference, we follow One-for-All [75] and train a lightweight dataset classifier (achieving $\geq 95\%$ accuracy) to predict the source dataset for an input sample at test time (Sec. B.2).

**Evaluation and Implementation Details.** To ensure fair comparison, we re-benchmark PTv3 under our unified protocol by varying batch composition and normalization, which yields an average improvement of 22 mIoU (Sec. B.3). We also compare against PPT combined with One-for-All, which uses ground-truth dataset labels during training and predicted dataset labels during inference. Model capacity is denoted with -S (small) and -L (large) suffixes for all architectures, including PTv3, Point-MoE, and PPT. PTv3-S and PPT-S use 14 encoder and 8 decoder layers, while their -L variants use 18 encoder and 12 decoder layers; in contrast, both Point-MoE-S and -L have 14 encoder and 8 decoder layers, with -S using 4 experts per MoE layer and -L using 8. The expert hidden dimension is set to 1× the input dimension in Point-MoE-S and 2× in Point-MoE-L. For details, please refer to Tab. 10 in the supplementary material.

### 4.2 EXPERIMENTAL RESULTS

**Single-dataset training generalizes poorly.** We begin by evaluating the standard setting where models are trained and evaluated on individual datasets. Note that in our setup PTv3 is trained and evaluated with CLIP-guided classification in an open-set setting, consistent with the multi-dataset regime [82], whereas the original PTv3 paper reports results under standard closed-set softmax label classification [81]. This difference makes our scores not directly comparable to the numbers in the original PTv3 paper. As shown in Tab. 1a, PTv3 achieves strong results on seen datasets, but its performance drops significantly on zero-shot datasets. For instance, a PTv3 model trained on S3DIS achieves only 3.5% on Structured3D and 4.9% on ScanNet. While dataset-specific models are not expected to generalize across datasets or categories, these results underscore the brittleness of

Table 1: **Results on seen datasets.** Results are reported in mIoU. Point-MoE achieves the best average performance without using domain labels. Methods marked with * use dataset labels during training. Models trained on a single dataset but evaluated on unseen datasets are in gray. (a) Indoor-only joint training on ScanNet [19], Structured3D [95], and S3DIS [1]. (b) Joint training across indoor and outdoor datasets: nuScenes [9] and SemanticKITTI [5].

| **(a) Indoor-only joint training** | | | | | | | |
|---|---|---|---|---|---|---|---|
| Methods | Params | Activated | ScanNet | Structured3D | | S3DIS | Avg. |
| | | | Val | Val | Test | Area5 | |
| *Single-dataset Training* | | | | | | | |
| PTv3-ScanNet [81] | 46M | 46M | 75.0 | 19.8 | 20.9 | 8.8 | 31.1 |
| PTv3-Structured3D [81] | 46M | 46M | 21.2 | 81.5 | 79.4 | 7.1 | 47.3 |
| PTv3-S3DIS [81] | 46M | 46M | 4.9 | 3.5 | 5.3 | 67.6 | 20.3 |
| *Multi-dataset Joint Training* | | | | | | | |
| OA-CNNs [52] | 52M | 52M | 71.6 | 59.0 | 58.6 | 60.6 | 62.5 |
| PTv3-S [81] | 46M | 46M | 71.3 | 56.6 | 55.9 | 69.4 | 63.3 |
| PTv3-L [81] | 97M | 97M | 72.2 | 56.0 | 58.1 | 67.4 | 63.4 |
| PPT-S* [75, 82] | 47M | 47M | 74.7 | 64.2 | 65.5 | 64.7 | 67.3 |
| PPT-L* [75, 82] | 98M | 98M | 74.8 | 64.7 | 65.9 | 65.0 | 67.6 |
| Point-MoE-S | 59M | 52M | 74.5 | 69.0 | 69.9 | 68.0 | 70.4 |
| Point-MoE-L | 100M | 59M | **76.0** | **69.5** | **70.8** | **69.5** | **71.5** |

| **(b) Indoor and Outdoor joint training** | | | | | | | | |
|---|---|---|---|---|---|---|---|---|
| Methods | Params | Activated | ScanNet | Structured3D | | S3DIS | nuScenes | SemKITTI | Avg. |
| | | | Val | Val | Test | Area5 | Val | Val | |
| PTv3-L [81] | 97M | 97M | 74.6 | 64.1 | 63.9 | 68.0 | 69.6 | 62.9 | 67.2 |
| PPT-L* [75, 82] | 98M | 98M | 76.7 | 67.5 | 66.0 | 61.9 | 70.8 | **66.9** | 68.3 |
| Point-MoE-L | 100M | 59M | **77.4** | **69.5** | **70.2** | **68.1** | **72.9** | 66.4 | **70.8** |

Table 2: **Zero-shot results on unseen datasets.** Point-MoE consistently achieves the strongest generalization across unseen datasets in the zero-shot setting. Methods marked with * use domain labels during training.

| **(a) Indoor-only joint training** | | | |
|---|---|---|---|
| Method | Matterport3D | | Average |
| | Val | Test | |
| PTv3-L | 39.0 | 45.4 | 42.2 |
| PPT-L* | 27.8 | 26.3 | 27.1 |
| Point-MoE-L | **41.0** | **46.7** | **43.9** |

| **(b) Indoor and Outdoor joint training** | | | | | |
|---|---|---|---|---|---|
| Method | Matterport3D | | Waymo | | Average |
| | Val | Test | Val | Test | |
| PTv3-L | 39.1 | 45.9 | 21.9 | 23.2 | 32.5 |
| PPT-L* | 24.9 | 23.1 | 16.3 | 16.7 | 20.3 |
| Point-MoE-L | **41.3** | **49.6** | **23.7** | **25.3** | **35.0** |

single-dataset training. This motivates the need for multi-dataset joint training strategies that expose the model to diverse data and promote broader generalization.

**Point-MoE surpasses all baselines on seen datasets, without dataset labels during training.** Point-MoE dispenses with dataset identifiers during training yet yields consistent gains across indoor benchmarks (Tab. 1a); both Point-MoE-S and Point-MoE-L variants improve on ScanNet, Structured3D, and S3DIS, with Point-MoE-L attaining a new state-of-the-art average of 71.5 mIoU. Despite joint training over multiple datasets, Point-MoE matches or exceeds single-dataset specialists, indicating effective expert routing rather than overfitting to any one source, or significantly exhibiting the negative transfer effects noted in PPT [82]. In the more challenging indoor and outdoor joint training setting (Tab. 1b), Point-MoE maintains its advantage across datasets and, on average, surpasses PTv3-L by 3.55 mIoU and PPT-L by 2.45 mIoU, demonstrating that label-free expert routing scales to heterogeneous datasets while simplifying the training pipeline. Moreover, performance on the seen indoor datasets remains closely aligned between indoor-only training and indoor-outdoor joint training, suggesting that Point-MoE preserves seen datasets accuracy even when subsequently trained jointly with outdoor data. This generalization is achieved implicitly in the model, without requiring dataset labels for incoming samples or maintaining explicit per-dataset normalization parameters. Note, since we use a lightweight dataset classifier, PPT's performance differs from what is reported

Table 3: **Ablations of Point-MoE designs and hyperparameters.** We conduct a systematic ablation on key design and hyperparameter choices of Point-MoE across four indoor multi-domain datasets: ScanNet (Scan), Structured3D (S3D), S3DIS, and Matterport3D (Mat).

| $\alpha$ | Scan | S3D | S3DIS | Mat |
|---|---|---|---|---|
| 0 | **74.5** | **66.9** | **68.6** | **40.1** |
| $10^{-4}$ | 72.3 | 64.1 | 67.9 | 39.5 |
| $10^{-3}$ | 71.6 | 58.8 | 67.4 | 37.4 |

(a) **Auxiliary load balancing loss.** Removing the auxiliary loss consistently improves performance.

| top $k$ | Scan | S3D | S3DIS | Mat |
|---|---|---|---|---|
| top 1 | 74.4 | 64.8 | 65.5 | 39.0 |
| top 2 | **74.5** | **66.9** | **68.6** | **40.1** |
| top 3 | 73.8 | 62.4 | 67.8 | 39.7 |

(b) **Top-$k$ expert selection.** Best performance is achieved with $k=2$ experts across datasets.

| norm | Scan | S3D | S3DIS | Mat |
|---|---|---|---|---|
| BN. | **74.5** | **66.9** | **68.6** | **40.1** |
| LN. | 70.8 | 56.9 | 67.3 | 37.2 |
| RMS. | 45.3 | 36.5 | 51.1 | 24.6 |

(c) **Normalization.** BatchNorm yields strongest performance; RMSNorm degrades results.

| case | Scan | S3D | S3DIS | Mat |
|---|---|---|---|---|
| FFN | 72.6 | 66.1 | 66.9 | 39.2 |
| Proj. | **74.5** | **66.9** | **68.6** | **40.1** |

(d) **MoE position.** Replacing the projection layer performs better.

| act. | Scan | S3D | S3DIS | Mat |
|---|---|---|---|---|
| SiLU | 73.6 | 65.7 | 64.9 | **40.1** |
| ReLU | **74.5** | **66.9** | **68.6** | **40.1** |

(e) **Activation functions.** ReLU performs the best.

| case | Scan | S3D | S3DIS | Mat |
|---|---|---|---|---|
| 1H | 73.0 | **60.3** | **66.3** | 39.5 |
| 2H | **73.5** | 58.6 | 65.4 | **39.8** |

(f) **Expert width.** H indicates original feature dimension.

| shared | Scan | S3D | S3DIS | Mat |
|---|---|---|---|---|
| 0 | **74.5** | **66.9** | **68.6** | **40.1** |
| 1 | 73.6 | 62.3 | 66.6 | 39.1 |

(g) **Shared experts.** Not sharing experts yields better performance.

| num. | Scan | S3D | S3DIS | Mat |
|---|---|---|---|---|
| 4 | **74.5** | 66.9 | 68.5 | 40.1 |
| 8 | 73.6 | **68.7** | 68.5 | **40.4** |

(h) **Number of experts.** More experts improve performance.

| batch | Scan | S3D | S3DIS | Mat |
|---|---|---|---|---|
| 4 | 74.0 | 64.9 | 66.6 | 38.1 |
| 6 | **74.3** | **66.9** | **68.6** | **40.1** |

(i) **Batch size.** Larger batchsize improves performance.

in the original paper, where oracle dataset identifiers are provided at inference. Finally, because only a subset of experts is activated, Point-MoE achieves a favorable accuracy–compute trade-off, cutting compute by 30.9% (265.7 vs. 384.4 GFLOPs) and peak VRAM by 19.0% (33.3 vs. 41.1 GiB) relative to PPT-L (Tab. 5). We analyze the emergent dataset-awareness and routing behavior in Sec. 4.4.

**Point-MoE outperforms all baselines consistently across zero-shot datasets.** As shown in Tab. 2a and Tab. 2b, Point-MoE-L achieves the strongest zero shot performance, including on Matterport3D under indoor only training and on both Matterport3D and Waymo under indoor and outdoor training, without using dataset labels during training. We believe a key reason is that label free expert routing encourages specialization based on the underlying semantics and geometry of the input rather than on dataset identity, resulting in more stable behavior under distribution shift. By contrast, PPT-L relies on explicit dataset labels during training. While this provides useful supervision on seen datasets, it can also encourage the model to depend too heavily on dataset specific cues. As a result, when a dataset is held out and its label is unavailable or no longer aligned with the training distribution at test time, this dependence becomes brittle and can hurt generalization. Additional analysis of Point-MoE's zero shot adaptation is provided in Sec. 4.4.

## 4.3 MoE Design Ablation

We investigate the impact of MoE design choices and hyper-parameters choices in indoor multi-dataset 3D semantic segmentation. Results are presented in Tab. 3. Note we are not using precise evaluator so results are not directly comparable to Tab. 1. Please refer to details in Sec. B.3 in the supplementary material.

**Effect of load-balancing loss.** The auxiliary load-balancing loss is designed to encourage uniform expert usage and prevent expert collapse [42, 56, 78]. We ablate its effect by training models with varying strengths of this loss in Tab. 3a. Interestingly, we find that removing the auxiliary loss entirely leads to consistently better performance across datasets. In contrast, stronger auxiliary losses result in significant degradation. We hypothesize that this is due to the inherent imbalance in the distribution of samples across datasets in 3D point cloud datasets—a phenomenon also observed by ChartMoE [87].

**Effect of activated experts.** Tab. 3b compares the performance of configurations with 1, 2 and 3 activated experts, finding diminishing return when increasing number of activated experts. We find that activating two experts yields the best performance across datasets. This analysis suggests that merely increasing the number of activated experts does not guarantee improved performance.

**Effect of MoE position.** We compare placing MoE in the feed-forward network (FFN) versus in the attention output projection $W_o$, holding all other components fixed. Projection-MoE consistently outperforms FFN-MoE in Tab. 3d. We hypothesize that $W_o$ is the fusion point where multi-head outputs are recombined *before* the next normalization, so experts placed there can route on richer, head-aggregated signals that preserve cross-dataset geometry cues, while leaving $W_q, W_k, W_v$ shared and stable; in contrast, FFN experts operate on post-normalized activations where such dataset or geometry cues are more

Table 4: **Ablations for mixed-dataset training.** "Mix" indicates that each training batch combines samples from multiple datasets.

| Methods | Mix | Scan | S3D | S3DIS | Mat | Average |
|---|---|---|---|---|---|---|
| PTv3 | | 48.4 | **57.8** | 44.1 | 27.8 | 44.5 |
| PTv3 | ✓ | **67.7** | 45.9 | **62.5** | **34.9** | **52.8**$_{(+8.3)}$ |
| PPT | | 74.2 | 58.2 | 61.0 | 24.3 | 54.4 |
| PPT | ✓ | **74.4** | **61.4** | **62.1** | **31.1** | **57.3**$_{(+2.9)}$ |
| PT-MoE | | 54.1 | 60.9 | 40.6 | 28.0 | 45.9 |
| PT-MoE | ✓ | **74.5** | **66.9** | **68.6** | **40.1** | **62.5**$_{(+16.6)}$ |

attenuated. This empirical gain leads us to adopt projection-MoE as the default. Similar observations also appear in recent work: *e.g.* UMoE [89] finds that using MoE for the output (and value) projections yields strong compute savings with high performance.

**Effect of model scaling with MoE.** Tab. 3h compares Point-MoE with 4 and 8 experts with activated experts fixed at 2. Increasing the number of experts from 4 to 8 leads to improved performance on seen and zero-shot datasets. Tab. 3f ablates the expert width by varying the intermediate feature dimension and the results show a mixed trend: wider experts improve performance on ScanNet and Matterport3D, but degrade results on Structured3D and S3DIS. Finally, we investigate the effect of adding shared experts [20, 65, 73]. We find that disabling shared experts consistently leads to better accuracy and dataset specialization across all benchmarks, as reported in Tab. 3g.

**Effect of hyper-parameter choices.** We provide justification for our selected hyper-parameters. In Tab. 3c, BatchNorm [29] consistently outperforms other normalization methods [3, 91] in the multi-datasets setting. Tab. 3e shows that ReLU [50] slightly outperforms SiLU [23]. Additionally, Tab. 3i demonstrates that importance of batch size in MoE training. In general, larger batch sizes lead to improved performance across datasets.

**Mixed dataset training.** In joint multi-dataset training, forming each minibatch with samples drawn from multiple datasets is consistently beneficial. Although there is no explicit batch-level interaction module, samples from different datasets still influence one another through shared normalization statistics and, for Point-MoE, through competition over a common expert pool. As shown in Tab. 4, this *mixed-dataset training* strategy improves the average mIoU of all backbones: PTv3 +8.3, PPT +2.9, and Point-MoE +16.6 mIoU on average. For dense backbones such as PTv3 and PPT, mixing exposes the model to more diverse domain statistics within each optimization step, which appears to regularize training and reduce overfitting to dataset-specific feature distributions. To make this comparison fair for PPT, we re-implemented its normalization module so that mixed-dataset batching is supported. The effect is most pronounced for Point-MoE, where diverse minibatches promote expert specialization across datasets, stabilizing routing and improving generalization.

## 4.4 ANALYSIS OF MOE

**Dataset clustering emerges in Point-MoE.** Fig. 2 shows t-SNE visualizations [70] of encoder and decoder features from PTv3, PPT [82], and Point-MoE. PTv3 exhibits no clear separation in either space, indicating limited capacity to form dataset-specific representations. PPT, by contrast, produces sharply separated clusters at both encoder and decoder levels, reflecting its reliance on dataset labels during training. While this aids seen-datasets performance, such rigid partitioning may hinder generalization capabilities as highlighted in Tab. 2. Point-MoE displays a different pattern: encoder features remain mixed across datasets, supporting shared representation learning, while the decoder disentangles dataset-specific structures. This apparent division of labor suggests that Point-MoE performs implicit dataset inference, without requiring explicit labels. Crucially, on the zero-shot Matterport3D dataset, Point-MoE aligns samples with the most related seen-dataset clusters (*e.g.*, ScanNet), which likely contributes to its strong generalization. Overall, these results show that Point-MoE can self-organize its experts and adapt to novel environments by effectively transferring knowledge across diverse training datasets.

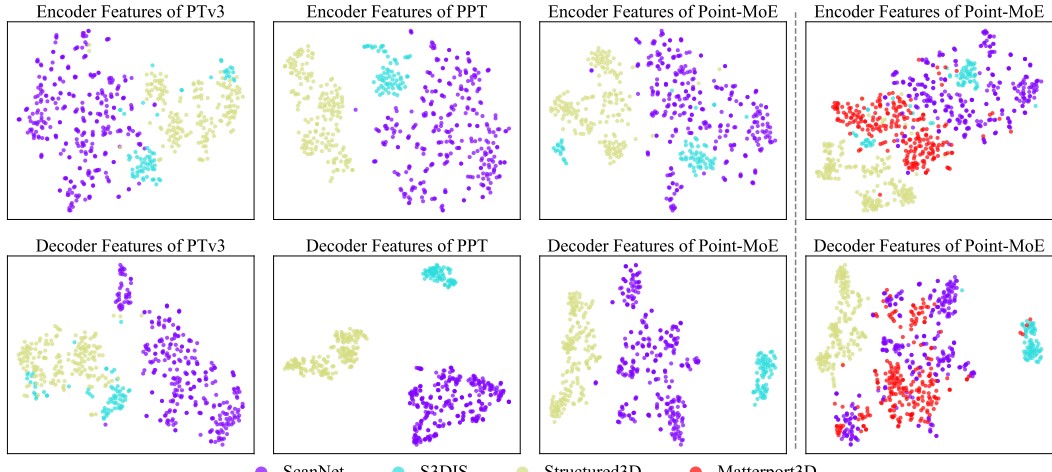

Figure 2: **t-SNE visualization of feature clustering.** The first three columns show that the decoders of both PPT and PointMoE have better separation between datasets in their decoder representations, than vanilla PTv3. The rightmost column highlights how Point-MoE generalizes to zero-shot datasets, with Matterport3D features aligning closely with ScanNet, which is indeed semantically similar.

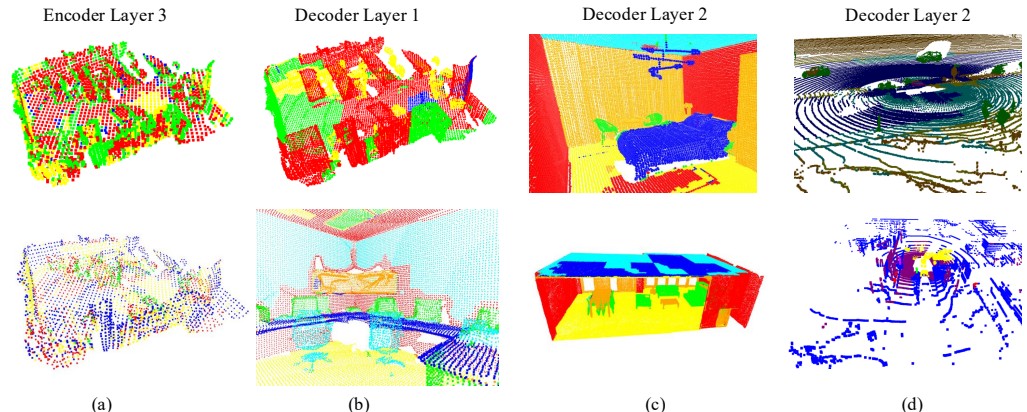

Figure 3: **Expert choice visualization.** Different colors within each image indicate different experts, showing that Point-MoE self-organizes point routing based on spatial and semantic cues across different layers. (a) shows one expert (green) focusing on edges, suggesting spatially aware mid-level routing in the encoder. (b) and (c) show that semantically related regions (*e.g.*, chairs and desks) are consistently assigned to the same expert, even across datasets. (d) shows meaningful routing in an outdoor scene with sparse LiDAR points despite lower point density and more irregular geometry.

**Expert choice visualization.** Fig. 3 visualizes expert assignments for validation scenes at selected encoder and decoder layers. In (a), the encoder layer primarily routes points according to local geometric structure. In particular, one expert (green) is consistently activated around object boundaries and thin structural regions, such as the edges of desks and chairs, while other experts dominate flatter surfaces, suggesting spatially aware mid-level routing in the encoder. In (b) and (c), the decoder layers exhibit more semantically coherent expert selection. Semantically related regions, such as chairs, desks, floors, and walls, are more consistently assigned to the same experts, even across different scenes and datasets, likely because decoder features are closer to the supervision signal and thus encode higher-level semantics. In (d), we examine an outdoor scene with sparse LiDAR points. Despite the lower point density and more irregular geometry, the model still produces meaningful routing patterns rather than random assignments. Nearby foreground regions and farther background regions tend to activate different experts, suggesting that the learned routing generalizes beyond dense indoor scans to more challenging outdoor environments. Additional visualizations are provided in Sec. C.3. We also observe a few isolated points assigned to different experts than their neighbors, possibly due to PTv3 design choices such as point serialization and positional encoding.

Table 5: **Inference efficiency.** FLOPs per step are measured with a profiler, and peak VRAM reports *allocated* memory. Relative deltas are computed with respect to PPT-L.

| Model | FLOPs/step (GFLOPs) | Peak VRAM (GiB) | Δ FLOPs vs. PPT-L | Δ VRAM vs. PPT-L |
|---|---|---|---|---|
| PPT-L | 384.4 | 41.1 | 0% | 0% |
| PTv3 | 375.8 | 37.4 | ↓ 2.2% | ↓ 9.0% |
| Point-MoE-L | 265.7 | 33.3 | ↓ 30.9% | ↓ 19.0% |

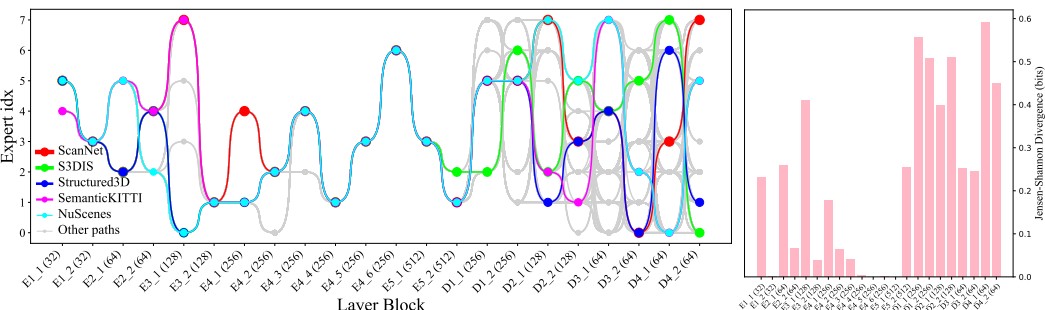

Figure 4: **Expert routing across datasets.** *Left*: the most frequent expert paths through each encoder (E) and decoder (D) layer, with channel sizes in parentheses, showing that expert selection varies across datasets. The effect is especially pronounced in decoder MoE layers. *Right*: Jensen Shannon Divergence (JSD) between dataset specific expert selection distributions at each MoE layer.

**Token pathways.** To understand how Point-MoE adapts to diverse datasets, we analyze expert routing at the token level. We track each token's top-1 expert across all MoE layers and reconstruct full routing trajectories from the final layer back to the first. We then rank expert paths by frequency and focus on the top 100 (Fig. 4). Encoder paths are notably less diverse than decoder paths, suggesting that encoders perform more dataset-agnostic processing. We also find sparse routing in deeper encoder layers, likely due not to feature reuse but to the U-Net design: features grow deeper spatially while tokens become sparser, reducing routing variability. When examining dataset-level trends, we find that certain dataset pairs, such as SemanticKITTI and nuScenes or ScanNet and Structured3D, share similar expert pathways, suggesting that Point-MoE implicitly clusters datasets with related geometric or semantic structures. To quantify these observations, we compute the Jensen-Shannon Divergence (JSD) [45] between expert selection distributions across datasets at each MoE layer. JSD [45] is an entropy-based measure of divergence between expert routing distributions across datasets, weighted by their token proportions; its formal definition is provided in the supplementary. As shown in Fig. 4 (right), decoder layers exhibit significantly higher JSD, indicating stronger dataset-specific specialization. Several encoder layers also display nontrivial JSD, underscoring the benefit of placing MoE throughout the network (see Sec. C.2 for more analysis).

**Efficiency analysis.** As shown in Tab. 5, Point-MoE-L requires only 265.7 GFLOPs per step and 33.3 GiB peak VRAM, reducing compute by 30.9% and memory by 19.0% relative to PPT-L (384.4 GFLOPs and 41.1 GiB). It is also more efficient than PTv3 (375.8 GFLOPs and 37.4 GiB). Despite this lower cost, Point-MoE-L achieves the best average mIoU in both settings, reaching 71.5 in indoor only joint training and 70.2 in indoor and outdoor joint training (Tab. 1).

## 5 DISCUSSIONS

This work embraces the "bitter lesson" [68] in AI: scalable generalization emerges from flexible architectures trained on diverse data, rather than hand-engineered domain priors. We introduce Point-MoE, a mixture-of-experts architecture for 3D understanding that dynamically routes tokens to specialized experts across both indoor and outdoor domains. Our experiments demonstrate that Point-MoE not only achieves superior performance on seen datasets but also generalizes effectively to zero-shot datasets without explicit domain labels. Through careful analysis of expert routing behavior, we show that the model self-organizes around both geometric and semantic structures, adapting its computation pathways to the nature of the input. These results highlight the potential of sparse, modular computation for robust and scalable 3D scene understanding. We provide more analysis of Point-MoE and discuss our limitations in the supplementary materials.

## 6 ACKNOWLEDGEMENT

The authors acknowledge the MathWorks Research Award, Adobe Research Gift, the University of Virginia Research Computing and Data Analytics Center, Advanced Micro Devices AI and HPC Cluster Program, Advanced Cyberinfrastructure Coordination Ecosystem: Services & Support (ACCESS) program, and National Artificial Intelligence Research Resource (NAIRR) Pilot for computational resources, including the Anvil supercomputer (National Science Foundation award OAC 2005632) at Purdue University and the Delta and DeltaAI advanced computing resources (National Science Foundation award OAC 2005572).

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
