APPENDIX

For a thorough understanding of our Point-MoE, we have compiled a detailed Appendix. The table of contents below offers a quick overview and will guide to specific sections of interest.

CONTENTS

# A   LIMITATIONS, LICENSES AND RISKS

## A.1   LIMITATIONS

Within our compute budget, we present a comprehensive study of MoE for point clouds—covering placement, expansion, routing choices (top-$k$, capacity factor, load balancing), and mixed-domain batching—with consistent gains. However, compute remains the primary bottleneck, capping Point-MoE's scale (experts, width, depth), training schedule, and number of jointly trained datasets. We therefore leave systematic scaling (e.g., expert count/width, per-stage sharing, alternative routers) to future work. Finally, because 3D semantic segmentation depends on costly human labels, we plan to pursue self-supervised pretraining on diverse unlabeled point clouds to reduce annotation effort and improve generalization. We hope these results serve as a simple, strong baseline and a useful reference point for future works.

## A.2   ARTIFACTS AND LICENSE

We report a list of licenses for all datasets and code used in our experiment in Tab. 6. We strictly follow all the datasets and code licenses and limit the scope of these datasets and code to academic research only.

## A.3   ETHICAL CONCERNS AND RISKS

This study does *not* involve human subjects or annotators. All experiments are performed on publicly released 3D benchmarks—ScanNet [19], Structured3D [95], S3DIS [1], Matterport3D [10], SemanticKITTI [5], nuScenes [9], and Waymo [67]—each distributed under their respective licences. Although these datasets are widely used, they may still reflect geographic, socioeconomic, or sensor–specific biases, which Point-MoE could inherit or amplify when deployed in real-world applications. We therefore encourage future work to develop debiasing techniques, to curate more balanced point-cloud corpora, and to incorporate fairness-oriented evaluation metrics so that cross-domain 3D models can serve all communities responsibly.

Table 6: **License information for the scientific artifacts used.**

| Data Sources | URL | License |
|---|---|---|
| ScanNet [19] | Link | ScanNet Terms of Use |
| Structured3D [95] | Link | Structured3D Terms of Use |
| S3DIS [1] | Link | CC BY-NC 4.0 |
| Matterport3D [10] | Link | Matterport3D License |
| nuScenes [9] | Link | CC BY-NC-SA 4.0 |
| SemanticKITTI [5] | Link | CC BY-NC-SA 3.0 |
| Waymo Open Dataset [74] | Link | Waymo Research License |
| Software Code | URL | License |
| Pointcept [18] | Link | MIT License |

Table 7: **Overview of 3D Semantic Segmentation datasets differences**. L, W, and H represent the length, width, and height. For all datasets, we use the raw point clouds to obtain the point ranges.

| Dataset | Sensor | Point Range (m) | Scene Type |
|---|---|---|---|
| ScanNet [19] | Reconstructed | L=[-2.13, 17.12], W=[-1.46, 18.19], H=[-0.38, 6.97] | indoor |
| S3DIS [1] | Reconstructed | L=[-37.93, 29.93], W=[-26.08, 46.06], H=[-2.65, 6.58] | indoor |
| Structured3D [95] | Synthetic | L=[-53.70, 52.05], W=[-82.07, 53.43], H=[-0.78, 15.49] | indoor |
| Matterport3D [10] | RGB-D Camera | L=[-34.99, 72.17], W=[-57.85, 75.65], H=[-6.79, 10.21] | indoor |
| SemanticKITTI [5] | 64-beam LiDAR | L=[-81.15, 80.00], W=[-80.00, 79.99], H=[-32.49, 3.51] | outdoor |
| Waymo [74] | 64-beam LiDAR | L=[-74.98, 76.53], W=[-75.00, 75.00], H=[-18.36, 6.78] | outdoor |
| nuScenes [9] | 32-beam LiDAR | L=[-105.17, 105.25], W=[-105.30, 105.17], H=[-53.42, 20.01] | outdoor |

## B ADDITIONAL EXPERIMENTS

### B.1 EXPERIMENT DATASETS

We compare a diverse set of 3D semantic segmentation datasets spanning both indoor and outdoor environments. ScanNet [19] is a large-scale indoor RGB-D dataset reconstructed from real-world scans of apartments and offices. S3DIS [1] contains real indoor scans of commercial buildings, providing richly annotated scenes with diverse architectural structures. Structured3D [95] is a synthetic dataset based on photo-realistic rendered scenes from CAD models, offering clean labels and dense annotations. Matterport3D [10] consists of real RGB-D scans captured in a wide range of residential environments, covering multiple floors and room types.

For outdoor settings, SemanticKITTI [5] provides point clouds captured by a 64-beam LiDAR mounted on a car, annotated for semantic understanding of dynamic street scenes. Waymo Open Dataset [74] extends this setup with more diverse driving environments and higher annotation quality, collected using a custom autonomous vehicle sensor suite. nuScenes [9] offers multimodal driving data, including LiDAR scans, captured in urban environments with complex traffic scenarios and longer temporal contexts. We include detailed dataset statistic in Tab. 7.

Furthermore, we provide detailed per-class comparisons on semantic classes for both indoor and outdoor datasets, as shown in Fig. 8 and Fig. 9.

### B.2 ADDITIONAL RESULTS AND ANALYSIS

**Domain Detector.** To enable domain-aware models at inference time—where ground-truth dataset labels are unavailable—we train a lightweight classifier to predict the domain of an input point cloud. This domain detector serves a purely functional role: it allows domain-conditioned variants of Point-MoE and other baselines to operate in realistic deployment scenarios where users may supply point clouds from unknown or mixed sources. During training, we observe that strong data augmentations (*e.g.*, grid dropout or random cropping) reduce the classifier's reliability by masking domain-specific cues. To preserve dataset-level characteristics critical for this task, we disable most augmentations and apply only minimal point cropping for efficiency. The domain detector converges quickly, with accuracy stabilizing after just a few epochs. As shown in Fig. 5, it achieves high

Table 8: **Categories of Indoor Datasets.**

| Dataset | #C | wall | floor | cabinet | bed | chair | sofa | table | door | window | bookshelf | bookcase | picture | counter | desk | shelves | curtain | dresser | pillow | mirror | ceiling | refrigerator | television | shower curtain | nightstand | toilet | sink | lamp | bathtub | garbagebin | board | beam | column | clutter | otherstructure | otherfurniture | otherprop | other |
|---|---|---|---|---|---|---|---|---|---|---|---|---|---|---|---|---|---|---|---|---|---|---|---|---|---|---|---|---|---|---|---|---|---|---|---|---|---|---|
| Structured3D | 25 | ✓ | ✓ | ✓ | ✓ | ✓ | ✓ | ✓ | ✓ | | | | ✓ | | ✓ | ✓ | ✓ | ✓ | ✓ | ✓ | ✓ | ✓ | ✓ | | ✓ | ✓ | ✓ | ✓ | | | | | | | ✓ | ✓ | ✓ | |
| ScanNet | 20 | ✓ | ✓ | ✓ | ✓ | ✓ | ✓ | ✓ | ✓ | ✓ | ✓ | | ✓ | ✓ | ✓ | | ✓ | | | | | ✓ | | ✓ | | ✓ | ✓ | | ✓ | | | | | | | ✓ | | |
| S3DIS | 13 | ✓ | ✓ | | | ✓ | ✓ | ✓ | ✓ | ✓ | | ✓ | | | | | | | | | ✓ | | | | | | | | | | ✓ | ✓ | ✓ | ✓ | | | | |
| Matterport3D | 21 | ✓ | ✓ | ✓ | ✓ | ✓ | ✓ | ✓ | ✓ | ✓ | | | ✓ | ✓ | ✓ | | ✓ | | | | ✓ | ✓ | | ✓ | | ✓ | ✓ | | ✓ | | | | | | | ✓ | | ✓ |

Table 9: **Categories of Outdoor Datasets.**

| Dataset | #C | car | bicycle | motorcycle | truck | other-vehicle | person | bicyclist | motorcyclist | road | parking | sidewalk | other-ground | building | fence | vegetation | trunk | terrain | pole | traffic-sign | barrier | bus | construction_vehicle | pedestrian | traffic_cone | trailer | driveable_surface | other_flat | manmade | other vehicle | sign | traffic light | construction cone | tree trunk | curb | lane marker | other ground | walkable |
|---|---|---|---|---|---|---|---|---|---|---|---|---|---|---|---|---|---|---|---|---|---|---|---|---|---|---|---|---|---|---|---|---|---|---|---|---|---|---|
| SemanticKITTI | 19 | ✓ | ✓ | ✓ | ✓ | ✓ | ✓ | ✓ | ✓ | ✓ | ✓ | ✓ | ✓ | ✓ | ✓ | ✓ | ✓ | ✓ | ✓ | ✓ | | | | | | | | | | | | | | | | | | |
| nuScenes | 16 | ✓ | ✓ | ✓ | ✓ | | | | | | | ✓ | | | | ✓ | | ✓ | | | ✓ | ✓ | ✓ | ✓ | ✓ | ✓ | ✓ | ✓ | ✓ | | | | | | | | | |
| Waymo | 22 | ✓ | ✓ | ✓ | ✓ | | ✓ | ✓ | ✓ | | | ✓ | | ✓ | | ✓ | | | ✓ | | | ✓ | ✓ | | | | | | | ✓ | ✓ | ✓ | ✓ | ✓ | ✓ | ✓ | ✓ | ✓ |

Table 10: **Training settings.** H indicates feature dimension.

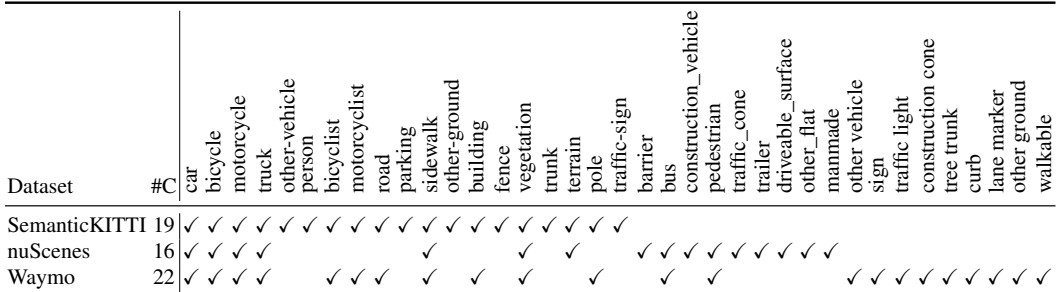

| Joint-training Indoor | | Joint-training Indoor-Outdoor | | Point-MoE-L | |
|---|---|---|---|---|---|
| **Config** | **Value** | **Config** | **Value** | **Config** | **Value** |
| optimizer | AdamW | optimizer | AdamW | # experts | 8 |
| scheduler | OneCycleLR | scheduler | OneCycleLR | topk | 2 |
| learning rate | 0.005 | learning rate | 0.005 | shared expert | 0 |
| weight decay | 0.05 | weight decay | 0.05 | activation func. | ReLU |
| batch size | 6 | batch size | 16 | auxiliary loss | 0 |
| datasets | ScanNet (1), S3DIS (1), Struct.3D (1) | datasets | ScanNet (1), S3DIS (1), Struct.3D (1), nuScenes (1), SemanticKITTI (1) | normalization | BatchNorm |
| | | | | domain-aware | False |
| | | | | domain rand. | False |
| iters | 140k | iters | 180k | expert capacity | 2H |

classification accuracy—99.2% for indoor-only and 95.5% for indoor-outdoor settings—making it suitable for practical use.

The high accuracy of the domain detector suggests that 3D datasets exhibit strong dataset-specific patterns, even when covering similar semantic categories. While we do not perform a detailed analysis of the underlying factors, differences in data acquisition, such as synthetic rendering, RGB-D reconstruction, or LiDAR scanning, are likely to introduce distinct geometric and distributional biases. These biases are implicitly learnable and can be exploited by simple classifiers. For conventional models, such domain discrepancies often necessitate careful data curation or normalization strategies to improve generalization. However, our findings with Point-MoE offer a different perspective: by explicitly modeling distributional variation through expert specialization, the architecture may naturally accommodate such domain shifts without requiring aggressive data homogenization.

**Does Mixture-of-Experts (MoE) work with a sparse convolutional backbone?** Yes. We integrate MoE into OA-CNN [52] by replacing the *SparseCNN* layer in the BasicBlock with a mixture of sparse-convolution experts (MoE-CNN), evaluating with top-2 routing. As summarized in Table 11, MoE improves over the joint-trained OA-CNN baseline on seen domains (ScanNet, Structured3D, S3DIS) and the unseen Matterport3D, raising the average from 57.3 to 60.1. This demonstrates that MoE is compatible with sparse convolutional backbones and delivers measurable gains, which further demonstrates MoE's applicability to architecture beyond transformers.

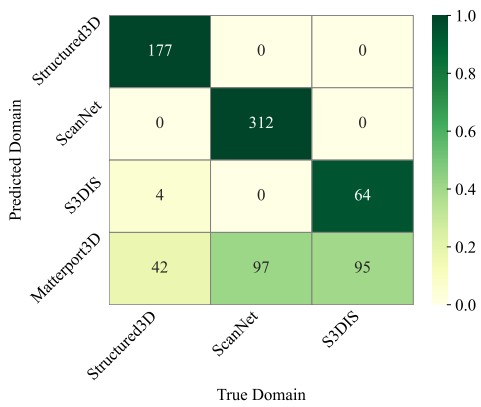 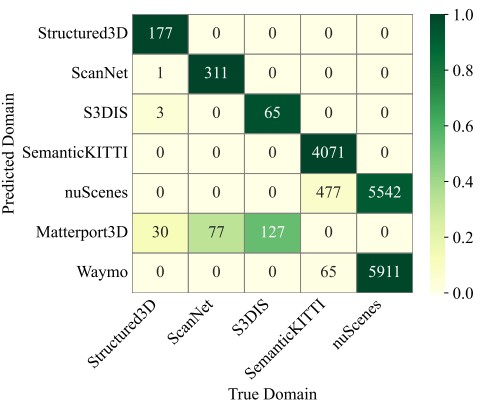

**(a)** Detector trained on indoor domains    **(b)** Detector trained on indoor-outdoor domains

Figure 5: **Domain Classification Confusion Matrices.** (a) Confusion matrix for a domain detector trained only on indoor datasets: Structured3D [95], ScanNet [19], and S3DIS [1]. (b) Confusion matrix for a detector trained on both indoor and outdoor datasets: Structured3D [95], ScanNet [19], S3DIS [1], SemanticKITTI [5], and nuScenes [9]. In both cases, we evaluate generalization to unseen domains. Values are normalized per predicted domain (row-wise).

Table 11: **MoE on a sparse convolutional backbone (OA-CNN).** Numbers are mIoU. Rows (1)–(4) use the public OA-CNN implementation (with a language-guided head for cross-dataset evaluation). (1)–(3) train on a single dataset; (4) joint-trains on ScanNet, Structured3D, and S3DIS; (5) replace SparseCNN with MoE-CNN using top-2 routing.

| Model | ScanNet | Structured3D | S3DIS | Matterport3D | Average |
|---|---|---|---|---|---|
| (1) OA-CNN-S3DIS | 5.70 | 3.8 | 65.3 | 5.4 | 20.6 |
| (2) OA-CNN-ScanNet | 72.6 | 21.6 | 8.4 | 37.4 | 35.0 |
| (3) OA-CNN-Structured3D | 20.8 | 65.6 | 6.8 | 18.9 | 28.0 |
| (4) OA-CNN | 71.6 | 59.0 | 60.6 | 37.8 | 57.3 |
| (5) OA-CNN-MoE | **72.0** | **65.3** | **63.0** | **40.2** | **60.1** |

## B.3 ADDITIONAL MAIN RESULTS

**Results without precise evaluator.** The results reported in Tab. 1 are obtained using the precise evaluator provided by Pointcept [18]. The precise evaluator first splits each scene into multiple overlapping chunks, then performs model inference on each chunk individually. After inference, it aggregates predictions across chunks through a voting mechanism to refine the final output. We typically observe a gain of 2–4% in accuracy on each dataset using this method. While effective, the precise evaluator introduces additional runtime overhead during evaluation. Moreover, we note that this form of ensembling may hurt overall performance when the model's predictions are unreliable in early chunks, as incorrect outputs are also accumulated in the final vote. To support convenient and reproducible benchmarking for the community, we include results with the precise evaluator in the main text and provide additional baseline results without it elsewhere in the paper.

From both Tab. 12 and Tab. 13, we observe that Point-MoE outperforms the baselines in most domains in both the indoor-only and indoor-outdoor joint training settings. These results are consistent with those obtained using the precise evaluator reported in the main text. We do not observe any contradictory trends, as the precise evaluator typically improves all methods by a similar margin of 2–4%. As such, our analysis and conclusions align closely with those presented in the main text. To ensure a strong baseline, we add mixed-dataset training and tune hyper-parameters, reporting the baseline with its best configuration. To be specifically, we apply mix-dataset training for baselines

Table 12: **Main Results without Precise Evaluator.** Each model is trained jointly on three indoor datasets and evaluated without precise evaluator. * indicates that each training batch combines samples from multiple datasets, methods with ** use mix dataset batch and layernorm [3].

| Methods | Params | Activated | ScanNet | Structured3D | S3DIS | Matterport3D | Average |
| | | | Val | Val | Area5 | Val | |
|---|---|---|---|---|---|---|---|
| *Single-dataset Training* | | | | | | | |
| PTv3-ScanNet [81] | 46M | 46M | 74.4 | 19.4 | 8.5 | 37.9 | 35.1 |
| PTv3-Structured3D [81] | 46M | 46M | 20.0 | 78.0 | 7.0 | 19.5 | 31.1 |
| PTv3-S3DIS [81] | 46M | 46M | 4.9 | 3.4 | 66.1 | 5.2 | 19.9 |
| PTv3-Matterport3D [81] | 46M | 46M | 52.7 | 23.0 | 9.5 | 53.6 | 34.7 |
| *Multi-dataset Joint Training* | | | | | | | |
| PTv3-S [81] | 46M | 46M | 48.4 | 57.8 | 44.1 | 27.8 | 44.5 |
| PTv3-S* [81] | 46M | 46M | 67.7 | 45.9 | 62.5 | 34.9 | 52.8 |
| PTv3-S** [81] | 46M | 46M | 70.0 | 56.0 | 66.9 | 35.6 | 57.1 |
| PTv3-L [81] | 97M | 97M | 57.1 | 58.1 | 40.3 | 31.0 | 46.6 |
| PTv3-L* [81] | 97M | 97M | 73.3 | 57.5 | 67.2 | 38.4 | 59.1 |
| PTv3-L** [81] | 97M | 97M | 74.0 | 67.4 | 67.8 | 40.6 | 62.5 |
| PPT-S [75, 82] | 47M | 47M | 74.2 | 58.2 | 61.0 | 24.3 | 54.4 |
| PPT-L [75, 82] | 98M | 98M | 74.7 | 64.9 | 62.0 | 27.8 | 57.4 |
| Point-MoE-S | 59M | 52M | 74.5 | 66.9 | **68.6** | 40.1 | 62.5 |
| Point-MoE-L | 100M | 60M | **75.1** | **69.3** | 68.5 | **41.3** | **63.6** |

Table 13: **Main Results on Indoor and Outdoor Scenes without Precise Evaluator.** Models are trained jointly on both indoor and outdoor datasets.* indicates that each training batch combines samples from multiple datasets, methods with ** use mix dataset batch and layernorm [3].

| Methods | Params | Activated | ScanNet | Structured3D | S3DIS | nuScenes | SemKITTI | Matterport3D | Waymo | Average |
| | | | Val | Val | Area5 | Val | Val | Val | Val | |
|---|---|---|---|---|---|---|---|---|---|---|
| PTv3-L [81] | 97M | 97M | 55.9 | 47.6 | 19.3 | 42.9 | 35.3 | 31.8 | 12.4 | 35.0 |
| PTv3-L* [81] | 97M | 97M | 73.4 | 61.9 | 61.4 | 64.7 | 59.3 | 38.9 | 22.8 | 54.6 |
| PTv3-L** [81] | 97M | 97M | 73.6 | 61.3 | 65.1 | 64.5 | 59.7 | 37.1 | 21.9 | 54.7 |
| PPT-L [75, 82] | 98M | 98M | 75.2 | 67.5 | 61.0 | 57.2 | **64.0** | 20.8 | 16.0 | 51.7 |
| Point-MoE-L | 100M | 60M | **76.2** | **70.1** | 67.9 | 68.9 | 63.3 | **40.7** | **23.4** | **58.6** |

we can directly apply without code modification and we also find layernorm is empirically better for the baselines.

# C  ADDITIONAL VISUALIZATION AND DISCUSSIONS ON POINT-MOE

## C.1  SEMANTIC CONCEPT DISTRIBUTION OVER EXPERTS

**Expert Specialization Word Cloud.** To visualize expert specialization, we collect the semantic predictions from each expert in the decoder layers of Point-MoE. During inference, we log which expert is selected for each point and record the corresponding semantic prediction. For each expert, we compute the frequency distribution of predicted classes across the entire validation set and display the most frequent class as a word cloud. Colors indicate the originating dataset of the prediction (*e.g.*, ScanNet, S3DIS, etc.), providing insight into cross-domain consistency.

We observe several interesting patterns in the expert specialization behavior. (1) In each decoder layer, there is typically at least one expert with a strong preference for outdoor data. For example, Decoder 0_0 Expert 7, Decoder 1_1 Expert 1, and Decoder 2_1 Expert 2 predominantly handle outdoor scenes. (2) Generic or ambiguous semantic classes such as *otherfurniture* or *other-ground* never dominate the assignment of any expert, suggesting that these classes are more diffusely distributed and do not concentrate through the routing mechanism.

From the visualization in Fig. 6, we observe several notable trends. Specialization emerges clearly: within each decoder block, experts tend to focus on semantically coherent regions such as *bed*, *bathtub*, *window*, or *trailer*, suggesting that experts develop functional roles under shared supervision. In many blocks, multiple experts cover similar classes, yet differ in dataset association, indicating potential

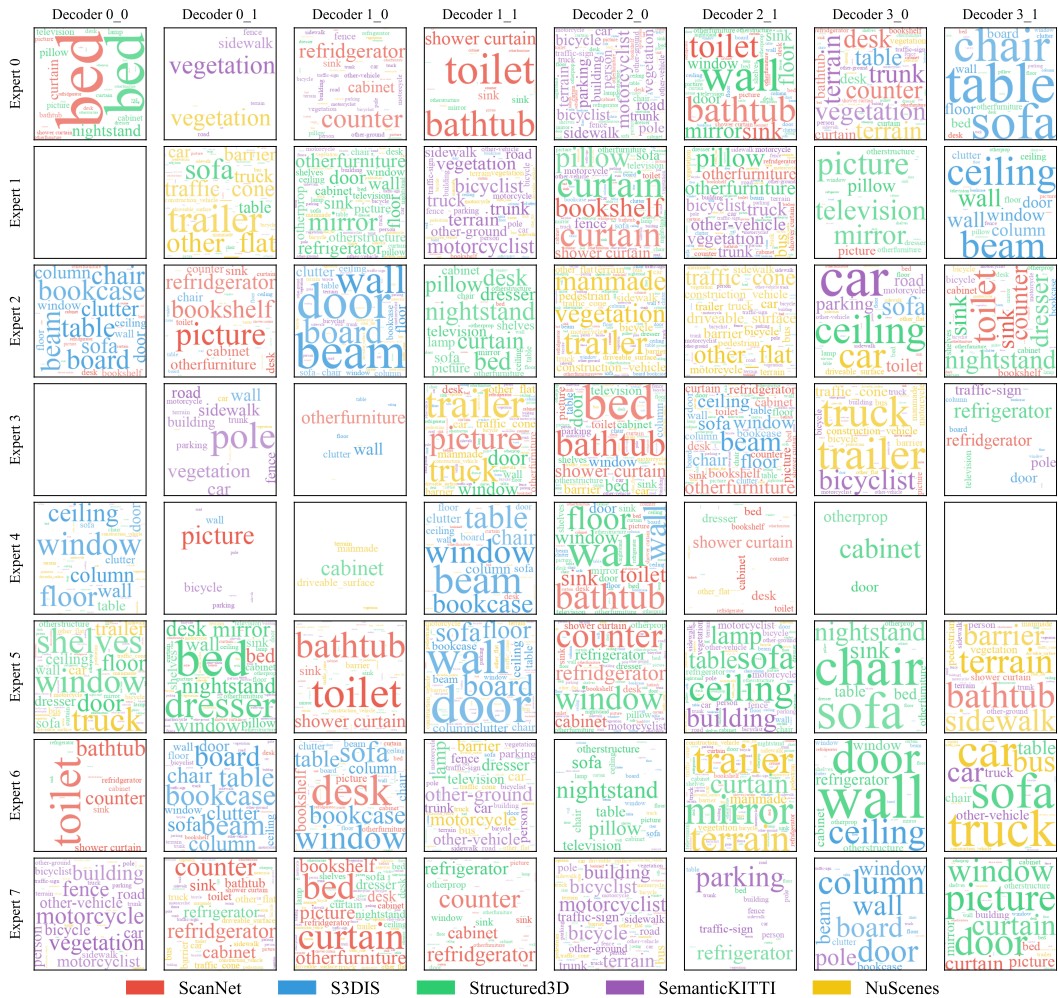

Figure 6: **Expert Specialization Word Cloud.** All classes' frequencies of selecting their top-1 expert are measured on the validation set. Colors indicate the originating dataset of the class (*e.g.*, ScanNet, S3DIS, etc.), providing insight into cross-domain consistency.

cross-domain generalization. For instance, certain experts across blocks frequently predict *bed* or *toilet* across multiple indoor datasets, hinting at robust feature preferences. We also observe signs of dataset sensitivity. Classes like *trailer*, *parking*, and *terrain* appear predominantly in SemanticKITTI and nuScenes, while *sink*, *shower curtain*, and *dresser* are more common in Structured3D. Comparing across decoder layers, earlier blocks display a broader mix of classes per expert, while deeper blocks exhibit more focused and object-specific specialization, reflecting an increasing semantic refinement through the decoding process.

These findings suggest that Point-MoE encourages structured representation learning, where each expert in a given layer implicitly captures recurring semantic or geometric patterns, even without explicit specialization objectives or layer-to-layer consistency.

**Expert Probability Stack Visualization.** To complement the word cloud analysis, we provide fine-grained views of expert behavior in Fig. 7, Fig. 8, Fig. 9 and Fig. 10. For each semantic class, we visualize the soft routing probabilities across all experts within a specific decoder block. Each horizontal bar corresponds to a class, segmented proportionally by the probability of top-1 expert assignment to each expert. This representation allows us to examine how responsibilities are distributed across experts for different categories.

We observe that certain semantic classes are strongly associated with a small number of experts. For example, *bed*, *bathtub*, and *toilet* are consistently routed to a narrow subset of experts, reflecting high specialization. In contrast, classes such as *wall* and *vegetation* are distributed across multiple experts, likely due to their spatial extent and ubiquity in scenes. Some experts show broad activation across diverse classes, while others are more selective, indicating a balance between general-purpose and task-specific routing. Several consistent patterns emerge in the decoder. In Decoder Layer 3 Block 0, both *truck* and *trailer* are primarily assigned to Expert 3, suggesting structural similarity. In Decoder Layer 3 Block 1, *dresser*, *nightstand*, and *counter* are all routed to Expert 2, reflecting shared visual or functional features. In Decoder Layer 0 Block 1, *traffic cone* and *barrier* exhibit nearly identical routing distributions, likely due to their geometric similarity and outdoor context. These observations confirm that the model learns structured yet flexible routing strategies, shaped by both semantic identity and spatial context.

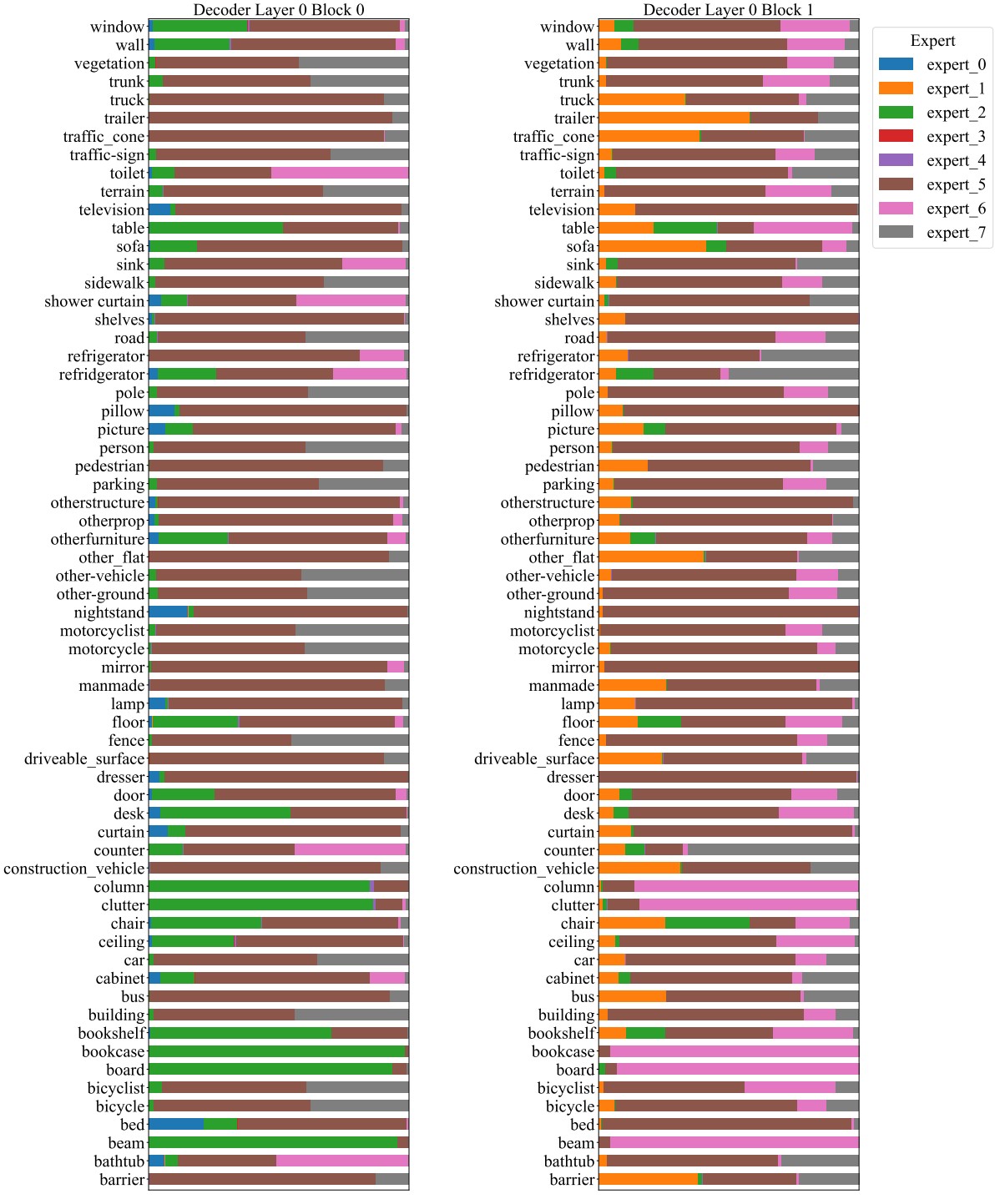

Figure 7: **Expert Choices Visualization for Decoder Layer 0.**

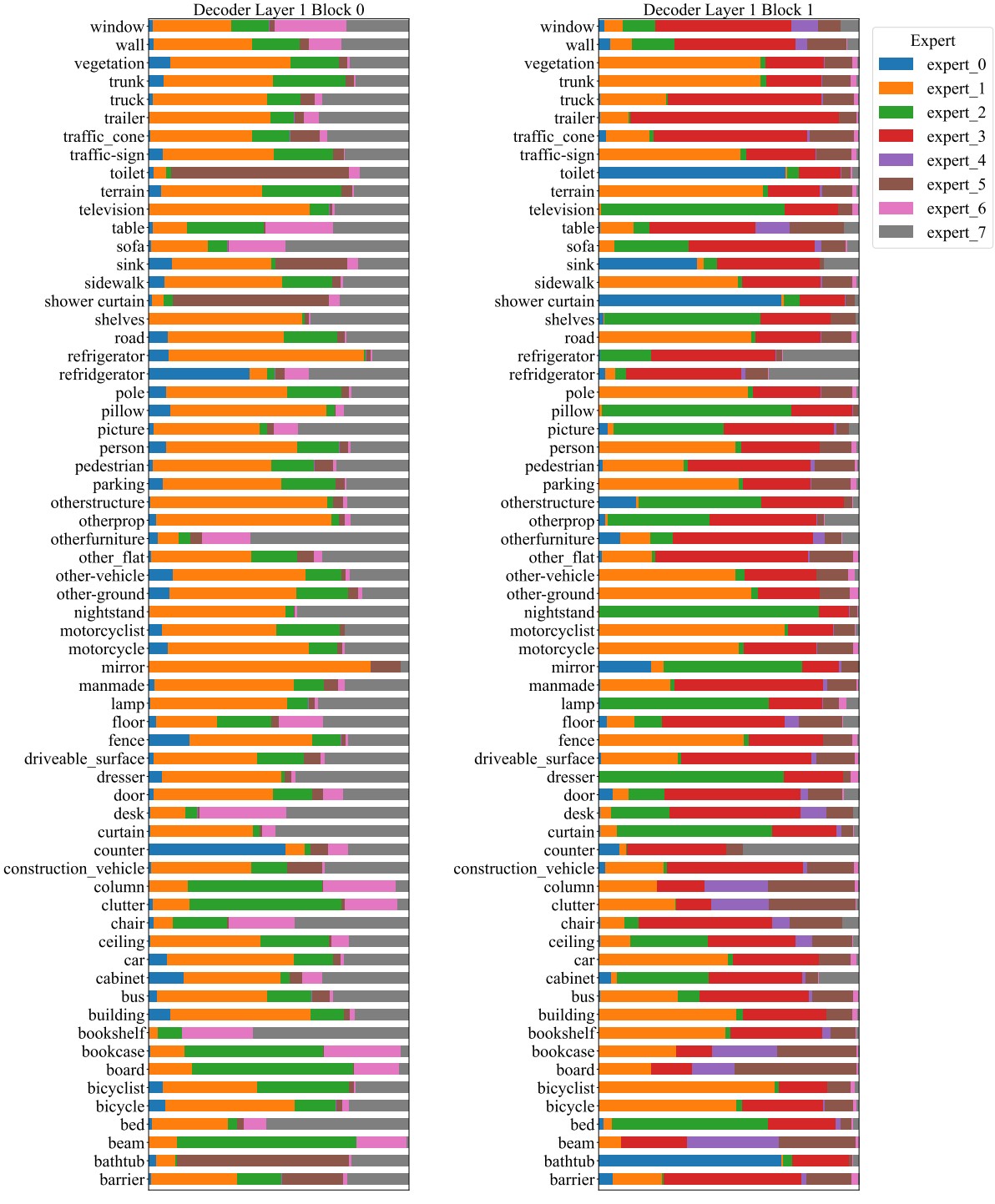

Figure 8: **Expert Choices Visualization for Decoder Layer 1.**

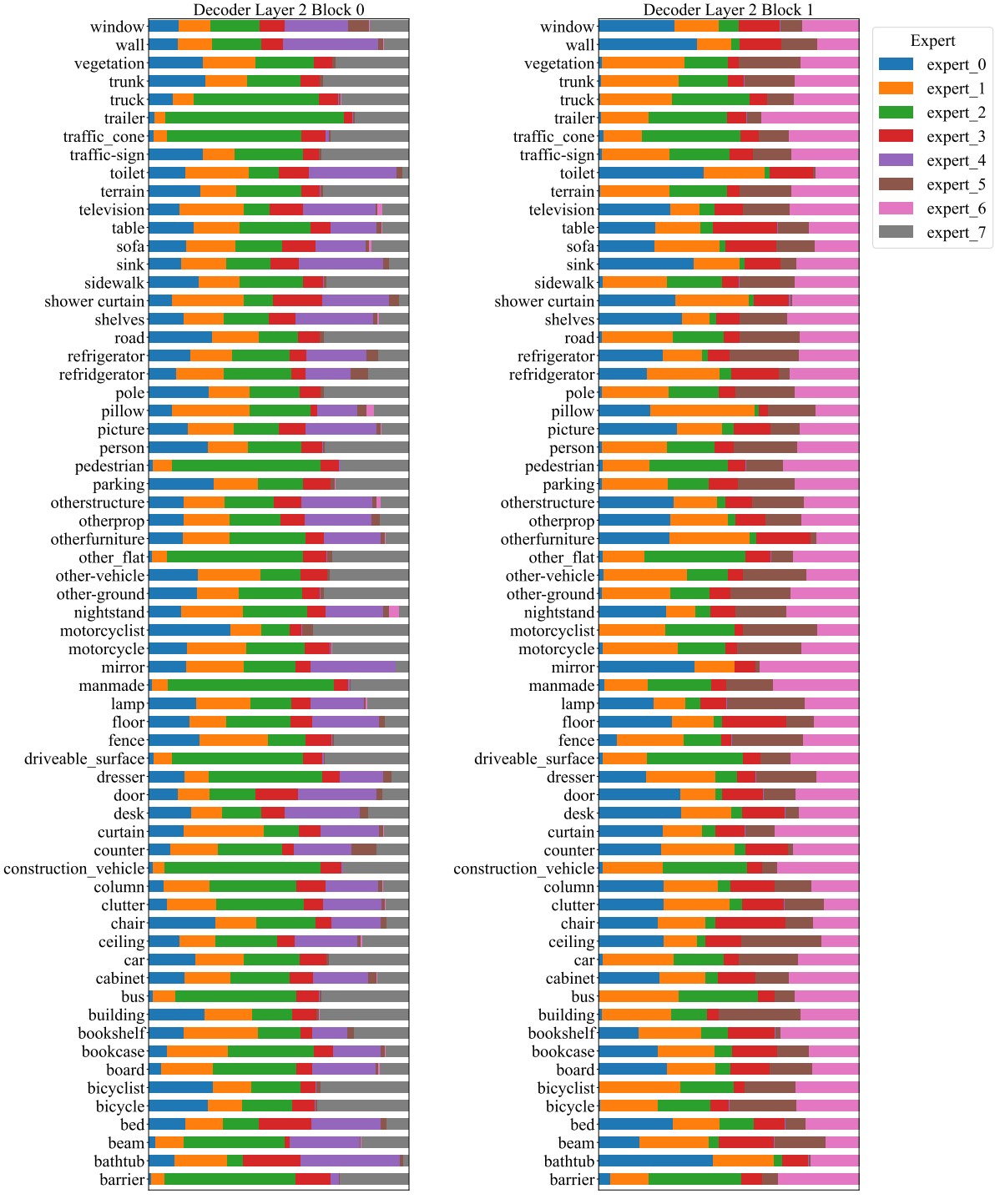

Figure 9: **Expert Choices Visualization for Decoder Layer 2.**

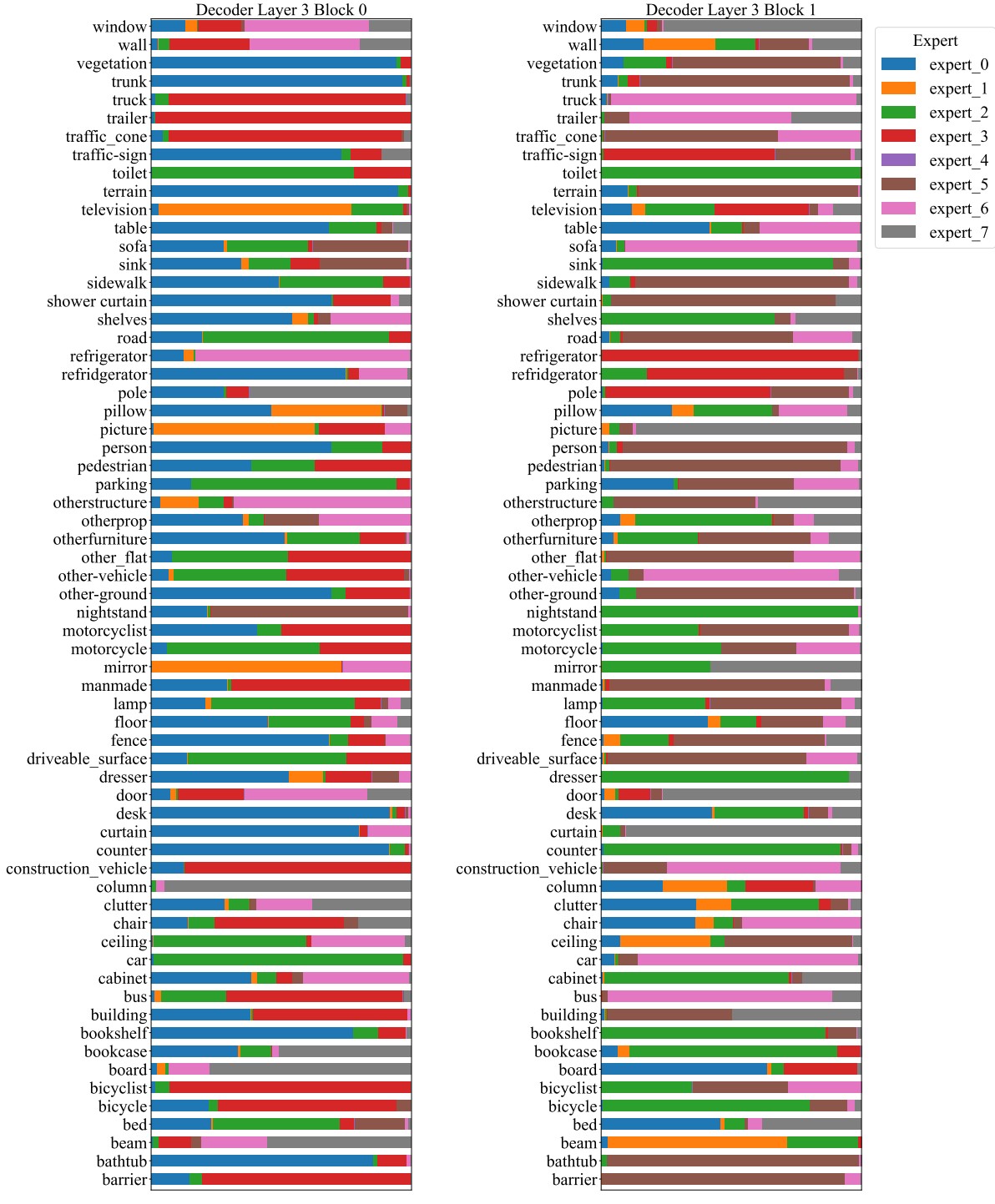

Figure 10: **Expert Choices Visualization for Decoder Layer 3.**

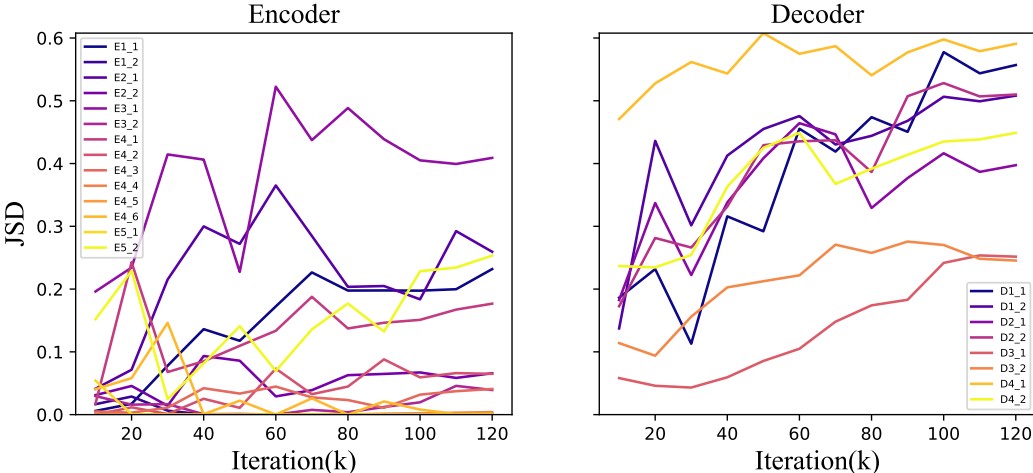

Figure 11: **JSD Dynamics During Training.** We visualize the Jensen-Shannon Divergence (JSD) between expert assignment distributions across datasets to study how expert specialization emerges over time. The plots illustrate JSD dynamics across training iterations for both encoder and decoder layers.

## C.2    EXPERT EVOLUTION.

In Fig. 11, we track the *Jensen-Shannon Divergence (JSD)* between expert assignment distributions across different datasets every 10k training iterations to analyze how expert specialization evolves over time. Let $P_1, P_2, \ldots, P_n$ denote the expert routing distributions for each of the $n$ datasets, where each $P_i$ is a categorical distribution over experts. Specifically, let $P_i(e)$ denote the fraction of tokens from dataset $i$ that are routed to expert $e$. Let $\pi_1, \pi_2, \ldots, \pi_n$ be the corresponding weights assigned to each dataset, satisfying $\sum_{i=1}^{n} \pi_i = 1$. These weights reflect the proportion of tokens contributed by each dataset. The Jensen-Shannon Divergence is defined as:

$$\text{JSD}_\pi(P_1, P_2, \ldots, P_n) = H\left(\sum_{i=1}^{n} \pi_i P_i\right) - \sum_{i=1}^{n} \pi_i H(P_i),$$

where $H(P) = -\sum_{x} P(x) \log P(x)$ denotes the Shannon entropy. This formulation captures the degree of divergence between datasets in terms of how they utilize the experts, providing a measure of specialization and separation over time.

In the early stages (10k–30k), JSD values remain low across most experts, indicating that the routing behavior is largely uniform across datasets and experts have not yet developed distinct roles. Between 40k and 60k iterations, we begin to observe divergence among certain experts, marking the onset of specialization. This trend continues and becomes more pronounced by 80k iterations, where several experts display increasingly distinct assignment distributions. By 100k iterations, the JSD curves stabilize, suggesting that expert roles have converged. Some experts diverge earlier and more sharply, indicating faster specialization, while others evolve more gradually. Overall, this analysis confirms that expert diversity in Point-MoE is not predefined but emerges progressively through training.

In the encoder, we observe that some experts maintain JSD values close to zero throughout training, suggesting that their routing behavior remains largely random and undifferentiated. This observation aligns with Fig. 4, where certain encoder experts do not exhibit clear dataset- or task-specific specialization. This raises the possibility that not all encoder layers require MoE modules, or that additional regularization or training signals may be needed to encourage expert differentiation in the encoder. Nonetheless, we do observe that some encoder experts do specialize over time, indicating that meaningful expert roles can still emerge under the right conditions.

In the decoder, we observe that all experts exhibit high JSD values, indicating that their routing behavior becomes highly specialized and dataset-dependent. Additional visualizations of expert assignments can be found in Sec. C.3, further illustrating this specialization.

### C.3 MORE EXPERT CHOICE VISUALIZATION

Fig. 12, Fig. 13, Fig. 14, Fig. 15, Fig. 16, Fig. 17, and Fig. 18 showcase expert assignments for a validation scene across all layers. Due to the hierarchical structure of the PTv3 backbone, we observe denser point cloud representations at the input and output stages, with sparser intermediate features. This design leads to visibly varying point densities and detail levels across the network. We also observe block-like artifacts in some visualizations, which are likely attributed to PTv3's point serialization strategy. In the encoder, certain experts occasionally exhibit collapsed or spatially clustered behavior—a phenomenon we refer to as "expert clasping." A deeper investigation into this behavior is left for future work. In the decoder layers, expert assignments closely resemble semantic segmentation outputs, even though the visualizations reflect only expert routing decisions rather than predicted labels. This suggests that experts naturally specialize in semantically coherent regions, despite the absence of explicit supervision guiding their roles.

In Fig. 12, we observe that early encoder layers primarily rely on geometric cues for expert routing. For example, certain experts—such as the green one—are consistently activated along object boundaries, including the edges of desks and chairs, while others tend to dominate on flat or horizontal surfaces. In the decoder layers, expert assignments become more semantically meaningful, with different experts focusing on specific object categories such as desks, chairs, walls, and floors.

In Fig. 13, we observe a similar overall pattern to ScanNet [19]. In the encoder, expert assignments are primarily driven by local geometric structure, with clear attention to furniture boundaries and wall intersections. Notably, edge-aware routing persists deeper into the encoder—for example, in encoder layer 2 block 0, certain experts remain focused on object contours. In the decoder, expert activation becomes increasingly aligned with semantic regions, such as tables, chairs, and structural elements. Despite the architectural and layout differences from ScanNet, the expert routing in S3DIS remains semantically consistent across the scene.

Fig. 14 illustrates expert routing on Structured3D [95], a synthetic dataset. Compared to real-world datasets, the expert assignment patterns appear sharper and more consistent, likely due to the absence of sensor noise and reconstruction artifacts. Some experts are clearly focused on flat regions, such as beds or floor. In the decoder layers, expert specialization aligns closely with object instances, and the separation across semantic regions is more distinct and well-defined, highlighting the benefits of clean, synthetic geometry for expert differentiation.

Fig. 15 presents expert routing on Matterport3D [10], which contains complex residential environments. In the early encoder layers, we observe clear expert usage patterns—for instance, a distinct separation between vertical and horizontal surfaces. In the decoder layers, expert assignments become more semantically meaningful, with experts specializing in large structural elements such as floors and ceilings, as well as common household objects like counters, sofas, and chairs. Notably, Matterport3D is *not* included in the training set, so this behavior reflects Point-MoE's ability to generalize expert specialization to *unseen* domains.

In Fig. 16, we examine an outdoor scene from SemanticKITTI [5], characterized by sparse LiDAR data. Despite the limited geometric density, the model still exhibits meaningful expert routing: nearby points tend to be assigned to certain experts, while distant points activate others. Although the transition from geometry-driven to semantically informed expert assignment remains observable, it appears noisier due to the inherent sparsity and irregularity of the input point cloud.

Fig. 17 shows expert routing on nuScenes [9], an outdoor dataset with lower-resolution LiDAR data. In the encoder layers, expert assignments appear more fragmented, likely due to the reduced point density. Nonetheless, we observe that closer points are often routed to specific experts, while distant points are handled by others, indicating a spatially aware routing pattern. In the decoder layers, expert assignments become more semantically coherent, with specialization emerging for vehicle bodies, building facades, and road surfaces based on point position and distance.

Fig. 18 presents expert routing on Waymo [67], an unseen dataset characterized by 64-beam LiDAR scans. In the encoder, experts exhibit structured activation patterns even in early layers, with clear delineation between flat surfaces and protruding objects. In the decoder layers, expert routing becomes more semantically aligned, with experts consistently activated over road surfaces, vehicle bodies, and trees. The assigned regions remain compact and well-separated.

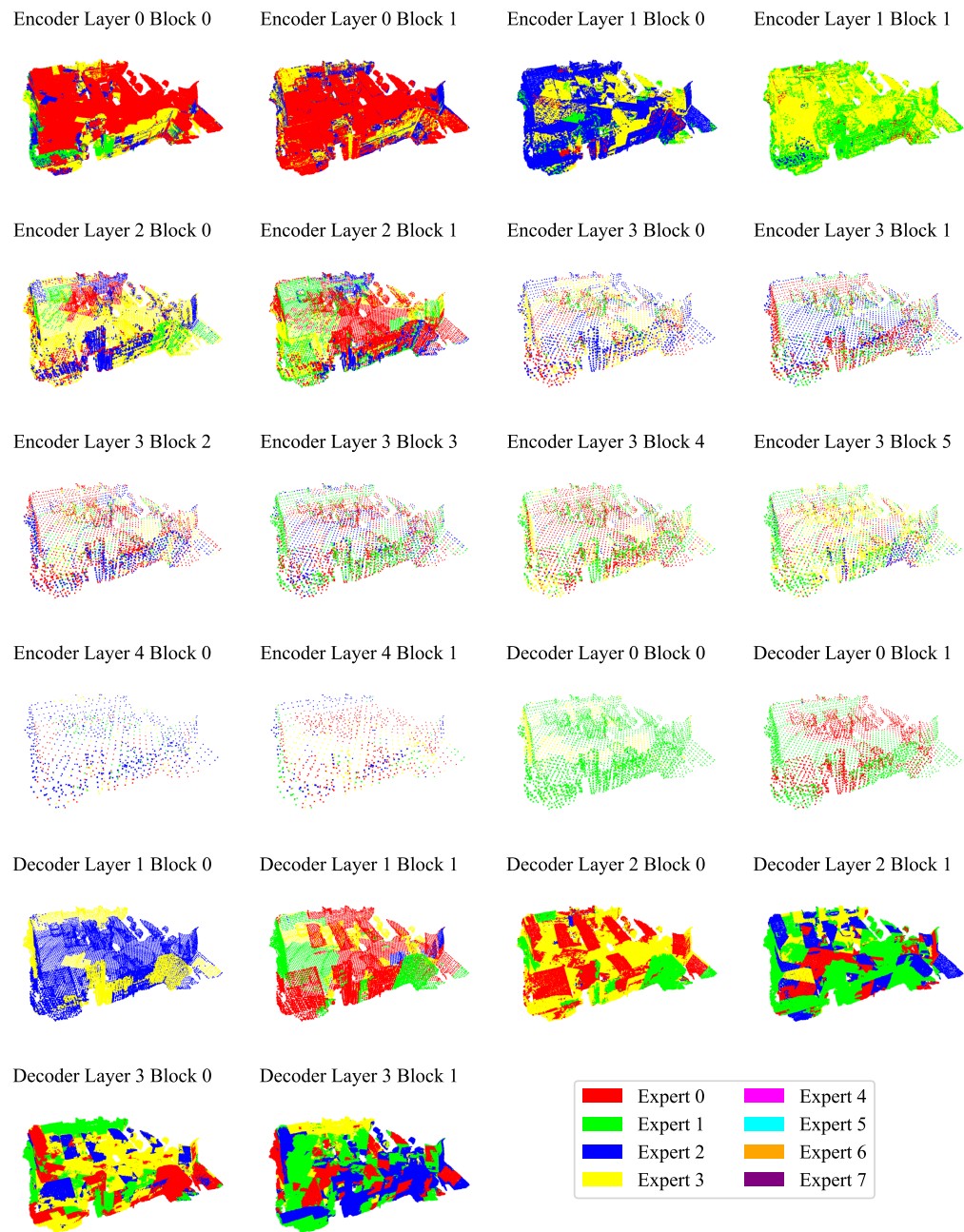

Figure 12: **Point-MoE's Expert Choice Visualization on ScanNet.** Each point is colored by its assigned expert across encoder and decoder blocks. Early layers show structured expert usage, while deeper layers exhibit more mixed routing, reflecting the evolving representation dynamics throughout the network.

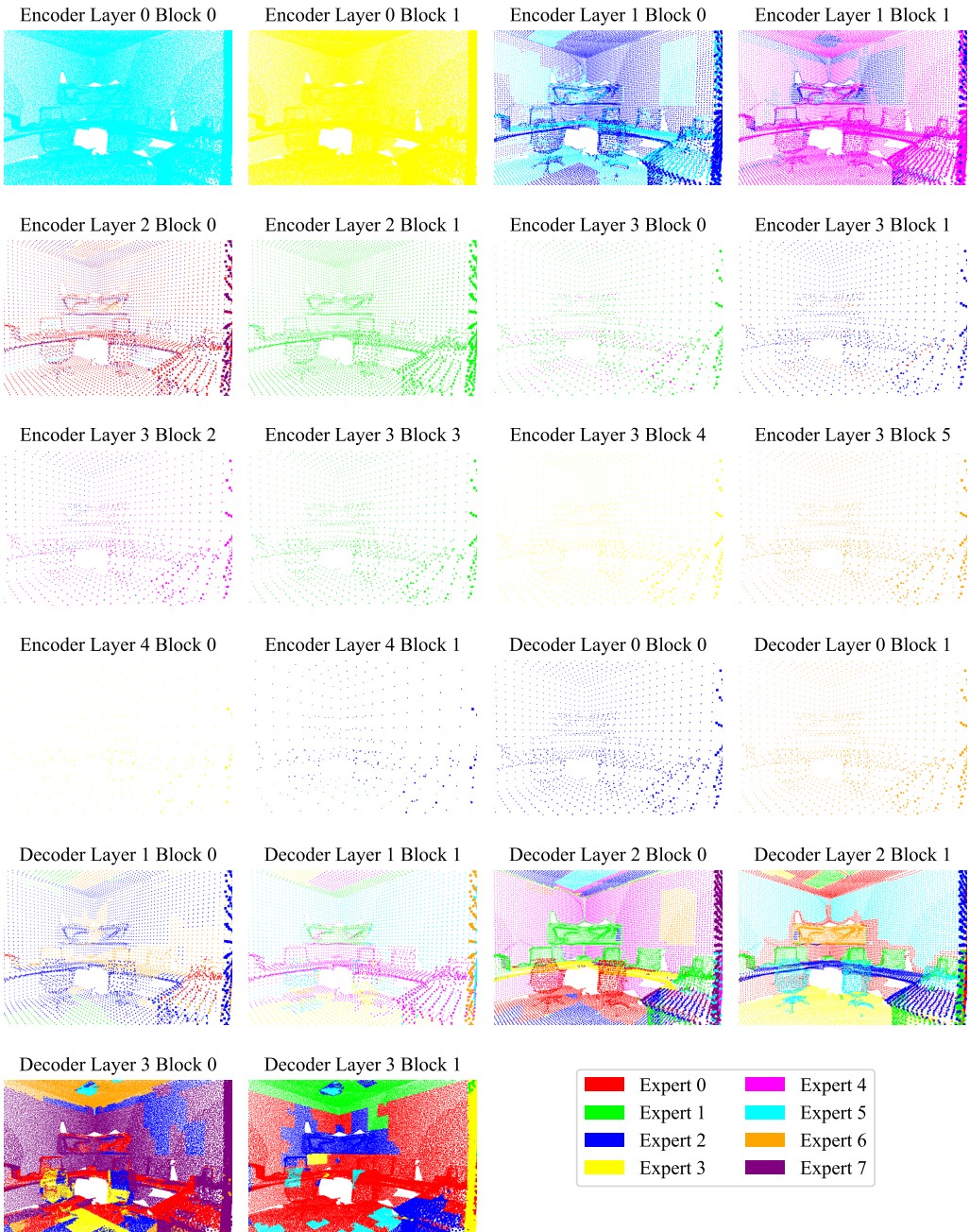

Figure 13: **Point-MoE's Expert Choice Visualization on S3DIS.** Each point is color-coded based on the expert it was routed to across both encoder and decoder blocks. The earlier layers demonstrate clear structure in expert assignments, whereas the later layers reveal more diverse routing patterns, indicating evolving feature representations.

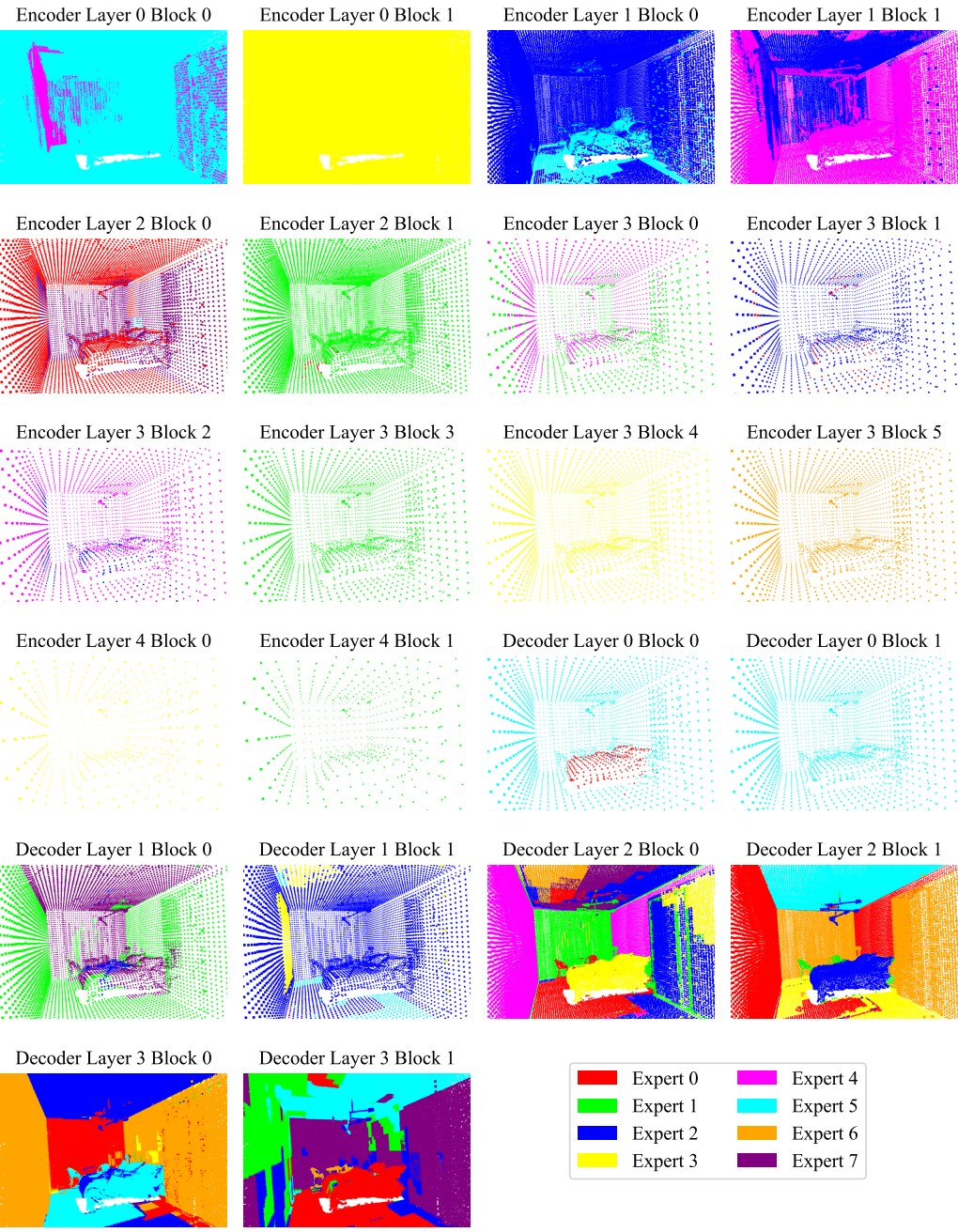

Figure 14: **Point-MoE's Expert Choice Visualization on Structured3D.** Points are colored by their designated experts in the encoder and decoder. While the initial layers display consistent expert allocation, the deeper layers show a blend of routing decisions, highlighting the network's shifting representational strategy.

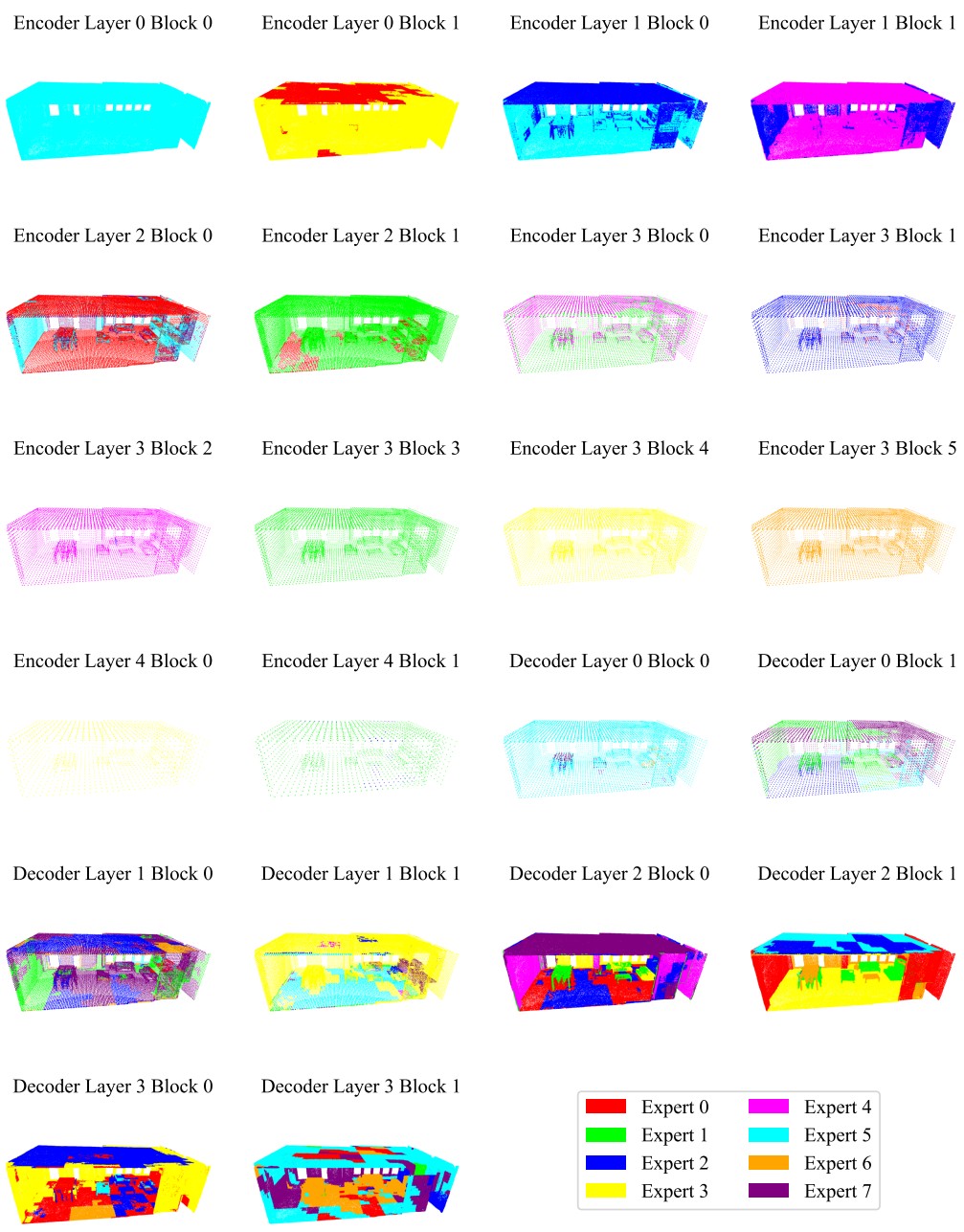

Figure 15: **Point-MoE's Expert Choice Visualization on Matterport3D.** Color represents the expert assigned to each point throughout the encoder and decoder. Structured expert usage is visible in early stages, transitioning to more varied routing in deeper layers, signaling changes in how features are processed.

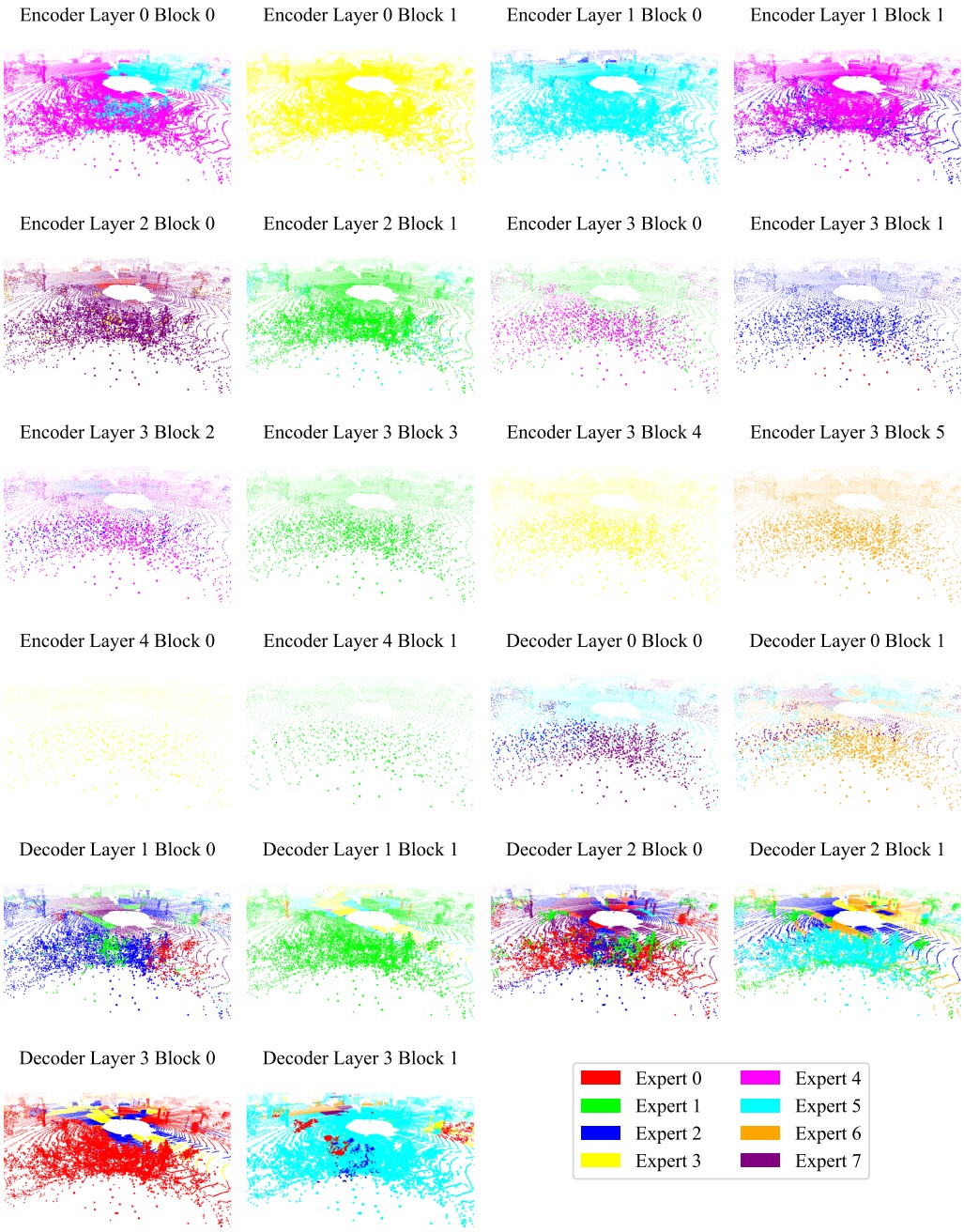

Figure 16: **Point-MoE's Expert Choice Visualization on SemanticKITTI.** The expert responsible for each point, shown through color, spans the encoder and decoder blocks. Initially, expert usage is highly organized, but becomes more dispersed in later layers, capturing the transformation in representation over depth.

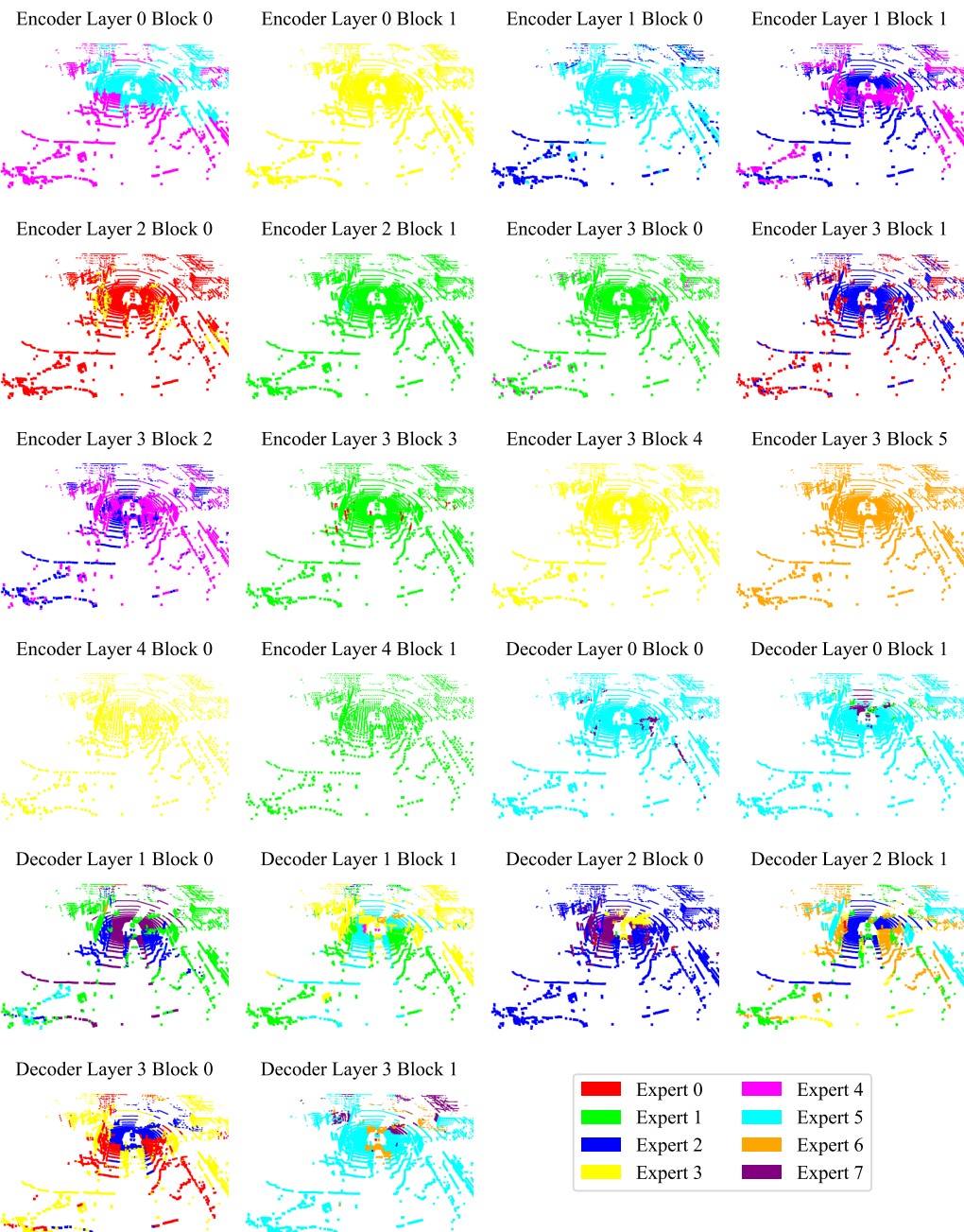

Figure 17: **Point-MoE's Expert Choice Visualization on NuScene.** Each point is colored according to its corresponding expert across the network's layers. Early encoder and decoder blocks exhibit distinct expert patterns, while deeper layers demonstrate a more distributed and dynamic routing behavior.

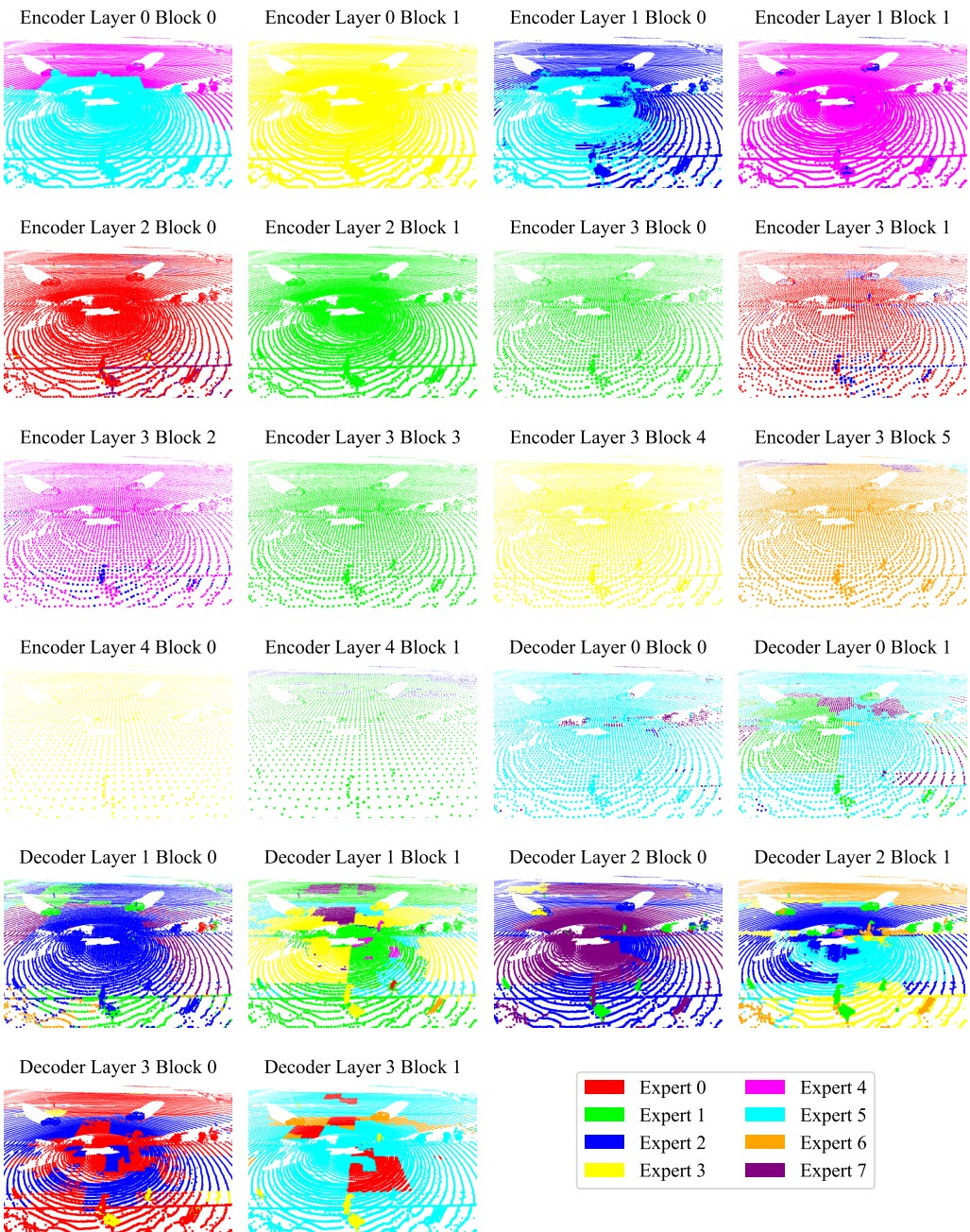

Figure 18: **Point-MoE's Expert Choice Visualization on Waymo.** Color indicates the expert each point is assigned to across encoder and decoder stages. Early network layers maintain structured expert routes, but this structure relaxes in deeper layers, illustrating evolving internal representations.