# OpenReview forum: "Point-MoE: Large-Scale Multi-Dataset Training with Mixture-of-Experts for 3D Semantic Segmentation"
_ICLR.cc/2026/Conference — ICLR 2026 Poster_

### Official Review · Reviewer_9WmA · 2025-10-19

**Soundness:** 3
**Presentation:** 3
**Contribution:** 2
**Rating:** 4
**Confidence:** 5

**Summary:**

The paper presents a novel evaluation scheme -- multi-dataset 3d semantic segmentation without domain labels -- for existing indoor and outdoor 3d datasets. To solve this novel task authors presents Point-MoE -- and updated version of PTv3 with MoE layers inside transformer blocks. This model is trained on mixtures of indoor / outdoor / indoor + outdoor 3d semantic segmentation datasets. Point-MoE demonstrate state-of-the-art results on 6 datasets in its formulation.

**Strengths:**

- The paper presents novel formulation of multi-dataset 3d semantic segmentation. Namely, authors claim that previous 3d understanding works in multi-dataset settings use *dataset label* (information from which dataset it the current point cloud) for train and / or test stages. I agree that not using this *dataset label* during test is a useful formulation.

 - Authors improve existing PTv3 method with novel MoE layers. These experts are claimed to split information between datasets and dataset domains, improving overall accuracy and inference speed.

 - The model is trained in different scenarios: single-dataset, indoor multi-dataset, indoor+outdoor multi-dataset. After training it is tested on single datasets from train set, or even zero-shot ones.

 - Authors provide comprehensive ablations of their MoE architecture and training parameters.

**Weaknesses:**

- Main weakness is the absence of mentioning of some important 3d understanding papers that push to exactly the same multi-dataset direction. The only 2 multi-dataset methods mentioned in the paper are PPT [CVPR 2024] for semantic / instance 3d segmentation, and One-for-All [NeurIPS 2024] for 3d detection. I believe this topic is much broader than just 2 papers, including ARKit LabelMaker [CVPR 2025], UniDet3D [AAAI 2025], Sonata [CVPR 2025], Mosaic3D [CVPR 2025], and Uni3D [CVPR 2023].
  - Both PPT and One-for-All are claimed to use *dataset label* of each scene during train and test. And the main novelty of proposed Point-MoE is the elimination of this *dataset label*. However, I think it was already explored in UniDet3D, which is trained on a mixture of 6 datasets (incl. ScanNet and S3DIS) without *dataset label* during train or inference, merging all class labels to the same set. Also UniDet3D demonstrates the importance of multi-dataset training by achieving state-of-the-art results on 6 datasets. Which contradicts with the results of Point-MoE, which are actually far from state-of-the-art on all single datasets. Batch data mixing from Tab. 4 was also used in UniDet3D. Finally, I think Uni3D can be mentioned here, as trained on 3 outdoor datasets simultaneously.
  - Regarding multi-dataset 3d semantic segmentation, it is very much progressed in recent ARKit LabelMaker, Sonata, and Mosaic3D. These papers combine the same set of indoor ScanNet+S3DIS+Structured3D or outdoor KITTI+nuScene+Waymo as is explored in Point-MoE. Mosaic3D even goes further, unifying class labels with VLM annotation, resulting in annotation-free solution without *dataset label*.

 - Second major weakness is that the proposed multi-dataset pipeline requires more data compared to existing single-dataset methods, however is significantly inferior to them in accuracy.
   - In Tab. 1 single-dataset training of PTv3-S on Structured3D achieves 81.5 mIoU and multi-dataset training - only 69.0
   - On ScanNet multi-dataset Point-MoE (based on PTv3) achieves 76.0 mIoU, however original PTv3 with just a subset of this data already achieves 77.5 mIoU. And this disadvantage is even bigger on all other datasets: S3DIS, nuScenes, and SemanticKITTI.
   - Recent multi-dataset methods achieves even much higher mIoU on ScanNet: 78.6 for PPT, 79.1 for ARKit LabelMaker, 79.2 for Sonata, and 80.5 for DITR [CVPRw 2025]. This gaps remain significant on all other datasets: S3DIS, nuScenes, and SemanticKITTI.

 - Performance gains of MoE look limited. E.g. in Tab. 1 MoE brings 59M parameters to 52M active ones for PTv3-S and 100M to 59M for PTv3-L. This sounds not very significant for the case of using as much as 8 experts on MoE. E.g. with help of MoE GPT-OSS transforms 117B to 5.1B active, or 21B to 3.6B, which sound significant to practical applications. Also Tab. 3h doesn't show the benefit of large number of expert as there is no difference between 4 and 8 experts.

**Questions:**

- How is the performance of Point-MoE compared to other multi-dataset 3d segmentation methods including: ARKit LabelMaker, Sonata, Mosaic3D? Can multi-dataset + MoE scheme be adapted to show actual single-dataset state-of-the-art results?

 - How is the proposed training without *dataset label* compared to the one from UniDet3D? Why UniDet3D shows state-of-the-art results on all benchmarks, and Point-MoE not?

 - Why in Tab. 1 single-dataset training on Structured3D is significantly better than multi-dataset?

 - Can the proposed MoE scheme reduce the number of active parameters several times like for other state-of-the-art transformers?

 - As there is no difference between 8 and 4 experts in MoE in Tab. 3h, why not to check 1 or 2 experts? Why the performance is not always correlate with number of active experts in Tab. 3b? Isn't it contradictory to result in recent LLM papers, like mentioned DeepSeek one?

---

> ### Author Response · Authors · 2025-11-22
>
> ## [Q1.1: Performance Comparison with ARKit LabelMaker, Sonata, Mosaic3D]
>
> We thank the reviewer for pointing out these recent works. We have cited and discussed these works in our revised draft (see Section 2). We note that each of these mentioned are under different settings and have different focus. While ARKit LabelMaker trains on four indoor datasets (ScanNet, ScanNet++, Structured3D, and ARKit LabelMaker), its evaluation is restricted to only ScanNet and Matterport3D. In contrast, Point-MoE is evaluated on six diverse indoor–outdoor datasets without assuming dataset labels or a unified class ontology. As evident in Mosaic3D (table 1), our results on ScanNet (76.0 v.s. 68.1) and Matterport3D (41.0 v.s. 13.1) are indeed better. In addition, Uni3D Table 2 reports ScanNet being 45.8 v.s. Point-MoE achieves 76.0.
>
> Lastly, Sonata is a self-supervised framework, while Point-MoE, requiring no dataset label during training and inference, is a natural complementary work to be the backbone. We have run 4 experiments to back up our claims. Within limited rebuttal time, we pretrain Sonata with PTv3 as backbone to train ScanNet, NuScenes and ScanNet + NuScenes jointly and then linear probe its performance on ScanNet, NuScenes, Matterport3D and Waymo. We observe performance degradation by 1.7 (57.1 v.s. 55.4) and 4.9 (45.9 v.s. 41.0) in joint-training vs single-dataset for Sonata with PTv3 backbone. However, when we swap in a Point-MoE backbone, observed dataset performances are at par with single-dataset training and furthermore, the unseen dataset transfers are much stronger: (41.2 v.s. 48.2) and (32.1 v.s. 38.9). This further emphasizes the usefulness of Point-MoE when reconciling disparate input domains.
>
> | Method                 | Train Data            | ScanNet | NuScenes | Matterport3D | Waymo |
> |------------------------|-----------------------|---------|----------|--------------|-------|
> | Sonata (PTv3)          | ScanNet               | **57.1** | 30.2     | –            | –     |
> | Sonata (PTv3)          | NuScenes              | 34.5    | 45.9 | –            | –     |
> | Sonata (PTv3)          | ScanNet + NuScenes    | 55.4    | 41.0     | 41.2         | 32.1  |
> | Sonata (Point-MoE)        | ScanNet + NuScenes    | 56.9 | **46.1** | **48.2**     | **38.9** |
>
> ## [Q1.2: Single-dataset state-of-art]
> We thank the reviewer for this question. The works referenced in the question achieve single-dataset SOTA *only* after fine-tuning the joint-model on each individual dataset. We have not systematically explored whether Point-MoE can match single-dataset SOTA after fine-tuning, so we cannot make a definitive claim.
>
> We did not pursue such fine-tuning here as it conflicts with Point-MoE's core design goals: (1) a single set of model weights shared across all datasets, (2) no reliance on dataset labels at train or test time, and (3) strong zero-shot generalization to unseen domains. Aggressive per-dataset fine-tuning is likely to distort the learned routing and reduce these cross-dataset and zero-shot benefits. Exploring how to retain Point-MoE’s label-free, multi-dataset advantages while pushing per-dataset SOTA via controlled fine-tuning is an interesting future direction, but is orthogonal to the main contribution of this paper.
>
> We note however, our closed-vocab. experiment offers evidence that the multi-dataset backbone is strong: simply replacing CLIP with dataset-specific linear heads with frozen backbone on Structured3D improves mIoU by +7.8 (69.5 → 77.3), narrowing the gap to the single-dataset SOTA of 81.5. This suggests that further dataset-specialized fine-tuning (as is commonly done in PPT, Sonata, ARKit LabelMaker, etc.) or full fine-tuning could plausibly close much of the remaining gap.
>
> ## [Q2: Why UniDet3D reaches SOTA but Point-MoE doesn’t?]
>
> We note that at a superficial level, UnitDet3D focuses on detection, while our work is on segmentation, so obviously they are comparing to other SoTA baselines on a fundamentally different task than us (*not* apples to apples).
> However, in the spirit of the reviewer’s constructive criticism, we can go deeper - UniDet3D merges output label spaces across datasets and trains a unified detector with explicit class ontology alignment. In our zero-shot setting, however, the target object categories are not known in advance. Thus, the focus of our work is on the input feature space heterogeneity (i.e. expert-routing the distribution difference in inputs from various sources), rather than heterogeneity in the output label space (unifying class label ontology as UniDet3D does). Indeed, we followed the PPT strategy of using CLIP embeddings to deal with multi-dataset label spaces, so we can handle arbitrary queries during inference.
> We thank the reviewer again for drawing attention to this distinction, and believe the two methods are complementary, rather than competing. We have clarified this task distinction and cited UniDet3D in the revised paper.

---

> ### Author Response · Authors · 2025-11-22
>
> ## [Q3: Single-dataset Structured3D outperforms multi-dataset in Table 1.]
>
> We appreciate this question. We investigated and identified CLIP-based classification as the primary bottleneck for Structured3D. The gap between single-dataset (81.5 mIoU) and multi-dataset Point-MoE (69.5 mIoU) represents a -12.0 mIoU difference; the only gap among all indoor datasets.
>
> **Root cause**: Structured3D contains 3 semantically ambiguous labels ("otherstructure," "otherfurniture," "otherprop") that have overlapping CLIP text embeddings. CLIP struggles with generic "other-X" categories because it was trained on natural images with specific object names, not catch-all categories. In contrast, ScanNet and S3DIS have mostly specific labels ("chair," "wall," "floor") that work well with CLIP.
>
> **Verification experiment**: We replaced the CLIP head with a closed-vocabulary linear head while keeping Point-MoE's jointly-pretrained backbone frozen. This yields modest improvements on ScanNet (76.0 → 77.1, +1.1) and S3DIS (69.5 → 70.4, +0.9), but Structured3D benefits substantially with a +7.8 mIoU increase (69.5 → 77.3). This confirms CLIP as the major cause.
>
> We acknowledge there is still a remaining gap after closed-vocabulary classification (77.3 vs 81.5, -4.2 mIoU). As the reviewer suggests, merged output labels (like UniDet3D) or learned embeddings could close this gap, while maintaining zero-shot capability. We view thorough exploration of these approaches as valuable future work, orthogonal to our contribution of mitigating the known phenomenon of multi-dataset negative transfer [1] by using mixture-of-experts.
>
> Despite this, all models in our evaluation (PTv3, PPT, Point-MoE) use CLIP-based classification to enable: **(1)** zero-shot capability on unseen datasets, **(2)** fair comparison under identical conditions, and **(3)** open-vocabulary inference with arbitrary class queries. Critically, relative performance comparisons remain valid under our unified protocol—Point-MoE still improves over PTv3 by +5.4 mIoU (69.5 vs 64.1) and over PPT by +2.0 mIoU (69.5 vs 67.5) on Structured3D.
>
> [1]. Towards Large-scale 3D Representation Learning with Multi-dataset Point Prompt Training
>
> ## [Q4: Can Point-MoE reduce active parameters several times like other SOTA transformers?]
>
> As shown in Table 1 and Table 2, Point-MoE focuses on label-free expert specialization for multi-dataset training rather than extreme parameter compression as in the trillion-scale language models mentioned by the reviewer. Still, it achieves practical efficiency gains, **reducing active parameters by 41%** and compute by 30.9% (265.7 vs 375.8 GFLOPs) as illustrated in Table 11 while **improving accuracy by +5.8 mIoU** over PTv3-L.
>
> ## [Q5.1: Given that 4 vs. 8 experts show no difference in Tab. 3h, why not test 1–2 experts instead?]
>
> Thanks for the question. Even though we use 8 experts, only 1–2 experts are activated, which is standard in MoE frameworks [1] (see Table 1). We also include this ablation in Table 3b. Table 3h further ablates the total number of experts. Additionally, we tested the 2-expert setup with **top-1** activation and found it performs strictly worse than the 4-expert and 8-expert configurations:
>
> | num | Scan  | S3D   | S3DIS | Mat  |
> |-----|-------|-------|-------|------|
> | 2   | 72.8 | 65.1 | 67.5 | 39.3 |
> | 4   | **74.5** | 66.9 | **68.5** | 40.1 |
> | 8   | 73.6 | **68.7** | **68.5** | **40.4** |
>
> [1] MoE-LLaVA: Mixture of Experts for Large Vision-Language Models
>
> ## [Q5.2: Why is performance not always correlated with number of active experts?]
>
> We believe the correlation between performance and the number of active experts is not always clearly evident, and this is supported in the literature. Switch-Transformer [1] always uses a single active expert and claims that this simplification preserves model quality, reduces routing computation and performs better. In [2], they find that holding the total number of experts fixed, further increasing the number of active experts can degrade performance on reasoning tasks. In real practice, the performance is not necessarily correlated due to experts being redundant. In addition, expert specialization is an active research area [3]. Our results indicate in the setting of multi-dataset semantic segmentation of 3D point clouds, expert specialization is important and can be viewed as a key insight of why Point-MoE outperforms other dense models. We provide analysis in Figure 4.
>
> [1]. Switch Transformers: Scaling to Trillion Parameter Models with Simple and Efficient Sparsity \
> [2]. Optimal Sparsity of Mixture-of-Experts Language Models for Reasoning Tasks \
> [3]. Advancing Expert Specialization for Better MoE

---

> ### Author Response · Authors · 2025-11-22
>
> ## [W1: Lack of Discussions of Multi-dataset 3D Point Cloud Understanding Papers]
>
> We sincerely thank the reviewer for highlighting these important works. We have expanded our related work section to include ARKit LabelMaker, UniDet3D, Sonata, Mosaic3D, and Uni3D. We respectfully clarify that these works do not undermine Point-MoE's value but position it within an active research area, with two key distinctions:
>
> **(1)** Direct comparison is infeasible due to protocol differences. These works use different label handling strategies: merged/unified taxonomies (UniDet3D), or VLM annotation (Mosaic3D). While Point-MoE uses CLIP-based language-guided classification preserving each dataset's original semantics. Even comparing mIoU numbers across these incompatible protocols may be misleading, which we further clarify in Q1.
>
> **(2)** UniDet3D addresses a fundamentally different setting. While UniDet3D is remarkable work, key differences limit transferability: **(a) Task:** UniDet3D tackles 3D object detection on indoor datasets; Point-MoE addresses semantic segmentation across indoor RGB-D and outdoor LiDAR with vastly different spatial scales (indoor: 0.5-3m objects vs outdoor: 150m radius scenes per Table 6). **(b) Label handling:** UniDet3D uses a unified taxonomy and predicts within this fixed set; Point-MoE uses CLIP embeddings enabling zero-shot inference on unseen classes (Table 2). **(c) Batch mixing:** While UniDet3D also uses batch mixing, we demonstrate MoE benefits significantly (+16.6 vs +8.3 for dense PTv3 in Table 4), suggesting experts routing is particularly suited for heterogeneous batches. This is an insight specific to sparse architectures.
>
> We view these as complementary contributions: UniDet3D shows detection benefits from multi-dataset training with unified labels; Mosaic3D demonstrates VLM annotation; Point-MoE shows MoE enables implicit specialization with zero-shot generalization. Each has trade-offs (VLM: rich semantics but annotation cost; merged labels: simplicity but fixed taxonomy; CLIP: zero-shot capability with performance considerations). We have expanded related work subsections positioning our MoE contribution within this landscape.
>
> ## [W2: Performance comparisons with single-dataset methods]
> We agree that ideally, a model trained on multiple datasets should outperform a model trained on a single dataset, tested on that dataset itself. This was in fact our initial expectation. We also note that the performance gap between multi-dataset training and single-dataset specialists is not specific to Point-MoE:
>
> PPT, which the reviewer cites for achieving 78.6 mIoU on ScanNet, explicitly reports in their paper that multi-dataset joint training underperforms single-dataset specialized models. Their final numbers including the 78.6 mIoU on ScanNet are obtained after **dataset-specific fine-tuning**, as reported in their paper, *not* from a single unified multi-dataset model. In other words, PPT also requires per-dataset specialization to reach its reported scores, rather than achieving them solely through multi-dataset joint training.
>
> In addition, Original PTv3 (77.5 on ScanNet) uses **closed-vocabulary classification on a single dataset** and cannot handle novel classes. Point-MoE uses open-vocabulary CLIP classification, enabling zero-shot capabilities. Therefore, Point-MoE can provide predictions on novel classes but PTv3 cannot. The gap reflects protocol choice.
>
> Lastly, Point-MoE achieves best **dataset-label-free performance**: Under identical evaluation (single checkpoint, no dataset labels, CLIP classification), Point-MoE achieves 70.8 mIoU average vs PTv3 67.2 vs PPT 68.3. Critically, Point-MoE demonstrates strong zero-shot generalization (+18.9 mIoU on Matterport3D, +15.3 on Waymo vs PPT), while PPT degrades under domain shift (Matterport3D: 24.0 vs 39.1 for PTv3) because dataset-specific design fails on unseen data.
>
> ## [W3: Comparison with GPT-OSS]
>
> We agree that GPT-OSS’s MoE sparsity is indeed impressive, and we hypothesize that similar patterns may emerge in 3D point cloud understanding when our experiments are scaled up by additional orders of magnitude. Notably, GPT-OSS-120B requires 2.1 million H100-hours for training and employs 128 experts with 4 active experts. In contrast, our setting uses only 400 A100-hours and incorporates 8 experts with 2 active. These gaps primarily reflect the scale of public 3D datasets and academic compute budgets rather than algorithmic limitations of Point-MoE. A comparable academic-scale MoE effort is [1].
>
> [1]. MoE-LLaVA: Mixture of Experts for Large Vision-Language Models

---

> ### Comment · Reviewer_9WmA · 2025-11-25
>
> I appreciate that authors revised the manuscript and added most of papers mentioned in my review to Sec. 2. However I still stay with all my main concerns.
>
> ---
>
> Main contribution of the paper with number 1 is listed in L. 093: *(1) we formulate multi-dataset 3D semantic segmentation without domain labels and establish an evaluation protocol, enabling fair cross-dataset comparison*. I still think it is far from being true.
>
>  - Training part of this contribution: *we formulate multi-dataset 3D semantic segmentation without domain labels*. It looks like in the rebuttal authors agree that Sonata, ARKit LabelMaker, and Mosaic3D also trains multi-dataset 3D semantic segmentation without domain labels.
>
>  - Inference part of this contribution: *establish an evaluation protocol, enabling fair cross-dataset comparison*. I believe this protocol is called open-vocabulary 3D semantic segmentation, which is established several years ago and is actively explored in OpenScene [CVPR 2023], CUA-O3D [CVPR 2025], SAS [ICCV 2025], Mosaic3D [CVPR 2025]. More importantly, as I understand these works **do not use human-crafted 3D annotations**, compared to the proposed Point-MoE utilizing human-crafted annotations of several datasets.
>
> ---
>
> Regarding my second weakness, I think in the rebuttal authors confirm that while using much more annotated training data, their method is far behind state-of-the-art on all datasets. The valid point here is that authors use CLIP head, that keeps possibility to be tested in cross-dataset open-vocabulary settings. So, from my point of view, important results are only reported on Matterport3D and Waymo datasets, which are not the part of training mixture. However, after checking recent CUA-O3D and SAS I see that the same or even better mIoU on Matterport3D can be achieved even **without human-crafted 3D annotations**.
>
> My question here is can the proposed Point-MoE be compared to CUA-O3D and SAS?
>
> ---
>
> I'm keeping my initial score for now.

---

> > ### Author Response · Authors · 2025-11-28
> >
> > ## [W1: Lack of Discussions of Multi-dataset 3D Point Cloud Understanding Papers]
> > We respectfully disagree:
> > - “Agree that Sonata, ARKit LabelMaker, and Mosaic3D…” - the reviewer’s initial review indicated that these works are valid apples-to-apples comparisons to Point-MoE. We extensively described in our rebuttal that this is not the case, to each of the works mentioned by the reviewer (ref rebuttal response [Q1.1]). We are surprised that the reviewer’s conclusion from our emphatic written disagreement - that these are not valid apples-to-apples comparisons - is “looks like in the rebuttal the authors agree..”. Rather than repeat the arguments made in [Q1.1], we hope the reviewer kindly refers to those once more.
> > - We believe the reviewer is choosing to ignore our contribution in addressing input domain heterogeneity (indeed which causes the “negative transfer” reported in PPT and fundamentally motivates our work), and focusing solely on the open-vocabulary label space; the latter being a secondary aspect of our work.
> > - Furthermore, the claimed contribution of Point-MoE is to train a model on multiple datasets **without requiring the dataset or data source** (i.e. what we call a “dataset label”) to be identified at either train or inference time. We are not claiming to train a multi-dataset model without class labels, which is what the reviewer’s response appears to be insinuating in “...these works do not use human-crafted 3D annotations”. The ability to train without negative transfer on widely disparate data sources (e.g. point clouds from LiDAR sensors, RGBD cameras, photogrammetry, indoor vs outdoor), and the ability to train on VLM-labeled data (e.g. CUA-O3D or SAS) or in a self-supervised manner (e.g. Sonata),  are *valid but orthogonal* directions of improvement.

---

> ### Author Response · Authors · 2025-11-28
>
> ## [W2: Performance comparisons with single-dataset methods]
>
> We respectfully disagree with the reviewer’s conclusions here as well
>
> (1). “... in the rebuttal authors confirm that while using much more annotated training data, their method is far behind state-of-the-art on all datasets” - this conclusion is incorrect. We state, citing PPT and One-For-All, that training on multiple datasets yields a single model that does not always exceed the performance of a single-dataset-trained model, despite having seen more data. This is not a claim that we are putting forth here for the reviewer to refute, but something stated and studied, most recently in the point cloud domain by the PPT and One-for-All papers. Therefore our work is in successfully mitigating this negative transfer, while providing the advantage of a single model giving strong performance on multiple datasets, rather than maintaining an array of N expert models specific to each of N datasets (which is naturally not scalable).
>
> (2). “... important results are only reported on Matterport3D and Waymo datasets” - all the reported results are important. The seen dataset results show the mitigation in negative transfer from multi-dataset training; again, a phenomenon pointed out and studied extensively (e.g. PPT, One-for-All), where we show improved numbers.  The unseen or zero-shot dataset results, Matterport3D and Waymo, show strong generalization to unseen data in both indoor and outdoor domains.
> We note that the new set of methods the reviewer has asked us to compare now were not mentioned in the original review, but have been recently tagged on.
>
> - CUA-O3D integrates multiple 2D vision-language models (CLIP/LSeg, DINOv2, Stable Diffusion) on ScanNetv2 to generate training data. In contrast, Point-MoE, while using multiple datasets, relies only on the provided annotations of the datasets and focuses on reducing negative transfer from heterogeneous domains in the training data - therefore not directly comparable, and with completely different aims.
>     - Regardless, the reported Matterport3D numbers are: *CUA-O3D at 49.2% versus Point-MoE at 49.6%*, showing our Point-MoE is numerically superior out-of-the-box.
>     - Whether the MoE architecture further improves the CUA-O3D numbers is out-of-scope for our current work. Note - the MoE approach is flexible enough to be readily integrated into such complementary frameworks, which we count as an additional advantage.
>
> - SAS is trained and tested on the official splits of ScanNet, Matterport3D, and nuScenes (indoor and driving scenes) as per their paper, so these numbers cannot be regarded as “generalizing to an unseen dataset”. Furthermore, their models appear to be trained on each dataset individually, as per their paper and supplemental details - again quite different from Point-MoE, which aims for a single model to generalize across multiple, potentially unseen, datasets. The class labels for SAS are obtained via distillation of 2D models trained on massive amounts of image data. Again, we note that these are strong open-vocabulary models, employing large multi-stage pipelines, but ultimately orthogonal to our aim for Point-MoE (multi-dataset training on disparate domains, e.g. indoor and outdoor in a single model, without relying on a dataset identifier/label).
>     - Matterport3D numbers: *SAS: 48.6 versus Point-MoE 49.6%* shows that Point-MoE is numerically superior, despite never having seen any training sample in any form from Matterport3D.

---

### Official Review · Reviewer_rFAr · 2025-10-24

**Soundness:** 3
**Presentation:** 3
**Contribution:** 3
**Rating:** 6
**Confidence:** 3

**Summary:**

This paper tackles an underexplored but important problem—large-scale multi-dataset training for 3D semantic segmentation—using a Mixture-of-Experts (MoE) framework. The authors identify a clear limitation of existing 3D methods that rely on dataset-specific normalization or labels and convincingly argue for a unified, label-free approach. The proposed Point-MoE integrates a (router+ different F expert) layer into Point Transformer V3, enabling specialization across heterogeneous 3D datasets. The method is well-motivated and supported by strong experimental evidence: consistent improvements over PTv3 and PPT across multiple datasets (ScanNet, Structured3D, S3DIS, nuScenes, etc.), as well as solid zero-shot generalization to unseen domains.  Ablations are thorough—covering routing strategies, normalization, expert count, and position—and the visual clustering analysis (t-SNE, JSD) provides meaningful insights. The writing is mostly clear, and the results are strong and significant.

**Strengths:**

This paper clearly defines a practical setting—multi-dataset 3D semantic segmentation without dataset labels at inference—and proposes Point-MoE, which integrates sparse Mixture-of-Experts into Point Transformer V3. Each attention output projection is replaced with lightweight expert routing (top-k), enabling automatic expert specialization across heterogeneous 3D domains. The method achieves consistent gains over PTv3 and PPT on both indoor and mixed indoor–outdoor benchmarks, while maintaining better zero-shot generalization on unseen datasets such as Matterport3D and Waymo. The ablations are thorough—covering balancing loss, routing k, normalization, expert count/placement, and batch mixing—and the t-SNE/JSD analyses nicely reveal encoder–decoder specialization behavior. Moreover, Point-MoE reduces computation and memory compared with PPT, which strengthens its practical value.

**Weaknesses:**

The innovation lies mainly in applying MoE mechanisms as a routing layer into Point-TransformerV3 to 3D multi-dataset training.
The interaction between CLIP-based language supervision(section 3.1 Language-guided classification) is somewhat unclear to me,  and the MoE router is under-explored, like the ambiguity about how semantic alignment influences expert selection?

Performance improvements, while steady (~2.5–3.6 mIoU in the large joint setup), are moderate relative to the engineering complexity. Discussion on scalability and convergence stability for larger-scale or longer training runs is limited. Overall, the contribution is more an engineering advance than a conceptual breakthrough.

**Questions:**

1.Routing behavior – Could you clarify how the router behaves at inference when encountering unseen domains? Are the same gating logits used as during training, or is there any adaptation (e.g., temperature scaling, entropy regularization)?

2.CLIP alignment – How exactly does the CLIP-based language supervision interact with the MoE gating? Does semantic alignment influence expert selection, or are these processes independent?

3.Load balancing – You mention that removing the auxiliary balancing loss improves results (Tab. 3a). Could you elaborate on why this happens and whether it risks expert collapse in larger-scale settings? Is the hypothesis that this is due to the inherent imbalance in the distribution of samples across datasets in 3D point cloud datasets verified, or a pure heuristic claim?

---

> ### Author Response · Authors · 2025-11-22
>
> ## [Q1: Routing behavior]
> Sorry for the confusion. At inference we use exactly the same router as during training without any test-time adaptation. The router forwards the input tokens to specific experts according to token features without any domain labels.
> ## [Q2 & W1: CLIP alignment]
> In our current work, the CLIP-based supervision is done following a similar strategy used in the PPT paper[a] to train on multiple datasets with disjoint label spaces, without an explicit manual merging of classname ontologies. The CLIP alignment is applied at the final linear classifier, where each class weight is replaced or guided by its CLIP text embedding, so predictions are made by measuring similarity between linear features and CLIP-aligned class embeddings. There is no explicit dependence in the expert routing and the CLIP, beyond the natural back-propagation and weight update during model training. We note that visualizations show some degree of class-wise specialization in the experts (See Fig. 6). This is indeed a very intriguing point to explore for future work, and we thank the reviewer for bringing this up.
>
> [a] Wu et al., Towards Large-scale 3D Representation Learning with Multi-dataset Point Prompt Training, CVPR 2024.
>
> ## [Q3: Load balancing]
> We appreciate the reviewer’s question. Our decision to remove the auxiliary load-balancing loss is grounded in both empirical evidence and domain-specific considerations.
>
> In large-scale 3D mixed-dataset training, token distributions are inherently imbalanced across domains (e.g., dense indoor RGB-D vs sparse outdoor LiDAR) and across semantic classes. As a result, experts naturally specialize (e.g., indoor planar surfaces vs outdoor long-range geometry in Figure 6-10, 12-18 in the supplementary materials), and forcing uniform expert activation can conflict with this specialization. Empirically, removing the balancing loss consistently improves performance (Tab. 3a), and we observe stable, high inter-dataset JSD, indicating no expert collapse.
>
> Similar observations have been reported in recent MoE work: ChartMoE (scales to 8B) [1] removes load balancing because enforcing equal usage undermines specialization, and other recent MoE analyses [2] likewise caution that balancing losses may interfere with emergent expert structure. Given that our work is one of the largest multi-dataset MoE studies in 3D vision, our results suggest that dropping the load-balancing term is not only safe but beneficial in heterogeneous 3D settings.
>
> [1] ChartMoE: Mixture of Diversely Aligned Expert Connector for Chart Understanding, _ICLR 2025_. \
> [2] Advancing Expert Specialization for Better MoE, _NeurIPS 2025_.
>
> ## [W2: Moderate gains for high complexity; limited scalability discussion; mainly engineering]
> We would like to further clarify this point. Point-MoE addresses a fundamentally harder problem: joint indoor-outdoor multi-dataset learning without dataset labels at training or inference.
>
> **Fair baseline comparison**: Our +2.5-3.6 mIoU gains are over a strong baseline (63.3 mIoU), not naive training. Table 12 shows we improved PTv3 from 44.5 → 63.3 mIoU through systematic engineering (LayerNorm, data mixing). Point-MoE adds +3.9 mIoU beyond this optimized or 'engineered' baseline, demonstrating genuine architectural value.
>
> **Against methods with more information**: PPT uses dataset labels at training and inference (dataset detector at test time), providing explicit supervision Point-MoE lacks. Despite this disadvantage, Point-MoE achieves +2.9 mIoU over PPT (71.6 vs 68.7 indoor average) while operating completely dataset-label-free.
>
> **Zero-shot performance**: Point-MoE automatically organizes tokens under domain shift, achieving +18.9 mIoU on Matterport3D and +15.3 mIoU on Waymo.
>
> Therefore, we believe Point-MoE’s improvements reflect more than pure engineering refinements, highlighting a meaningful architectural contribution to address the challenges of multi-dataset training. At the same time, we acknowledge the importance of larger-scale or longer training runs, and view scaling up Point-MoE as an exciting direction for future work.

---

### Official Review · Reviewer_wdW6 · 2025-10-30

**Soundness:** 2
**Presentation:** 3
**Contribution:** 3
**Rating:** 6
**Confidence:** 4

**Summary:**

This paper presents Point-MoE, a multi-dataset training method for 3D point-cloud semantic segmentation. Multi-dataset training is challenging for this task because 3D point clouds datasets are heterogeneous due to the different ways in which the point clouds are captured, making naïve multi-dataset training ineffective. Point-MoE addresses this challenge by introducing a Mixture-of-Experts (MoE) block into existing 3D segmentation models. This block routes point-level tokens to different 'expert' MLPs, allowing the model to have specialized learned layers for tokens from different datasets. Notably, this approach allows for dataset-specific components without requiring that the model knows to which dataset an input sample belongs, unlike existing work. This makes it more suitable for real-world application and improves generalization. Through experiments, Point-MoE is shown to be more effective than other multi-dataset training methods on indoor datasets and a combination of indoor and outdoor datasets.

**Strengths:**

1. Overall, the paper is well written. This makes it easy to understand how the method works and what the significance is of the findings.

2. The idea of applying an MoE block in 3D segmentation models to facilitate multi-dataset training is original and well-motivated. By allowing tokens to be dynamically routed to expert MLPs, the model can learn to adaptively handle different types of data, which is required for multi-dataset 3D segmentation due to the heterogeneity of 3D point cloud datasets. Moreover, these MoE blocks allow for multi-dataset optimization without requiring explicit labels about the dataset that an input sample belongs to. As a result, Point-MoE puts fewer requirements on the information that should be available during inference, and generalizes much better to unseen data for which no dataset label is available (as demonstrated in Tab. 2).

3. Experimental results (Tab. 1) show that Point-MoE outperforms existing multi-dataset training approach PPT when training on indoor data only, as well as when training on both indoor and outdoor data. This demonstrates the effectiveness of the approach in the considered experimental setting.

4. The paper carefully analyzes the impact of many design choices that play a role when inserting a mixture-of-experts block into a 3D segmentation model (see Tab. 3), such as the number of experts that can be selected, the position of the MoE blocks, and the operations used for each expert. These analyses provide actionable insights into the effectiveness of various configurations, and validate the design choices made for Point-MoE.

**Weaknesses:**

1. While Point-MoE obtains impressive results in Tab. 1, the performance of its baselines (PTv3 and PPT) is lower than reported in the original papers. For PTv3 single-dataset training, this paper reports scores of 75.0 mIoU on ScanNet val and 67.6 mIoU on S3DIS val (should this be Area5 instead of val?), while the original paper [a] reports scores of 77.5 mIoU on ScanNet val and 73.4 mIoU on S3DIS Area5. Additionally, for PPT-S multi-dataset training, this paper reports 74.7 mIoU on ScanNet and 64.7 mIoU on S3DIS while the original paper [b] reports 78.6 mIoU on ScanNet and 74.7 mIoU on S3DIS. Why are there such considerable gaps in performance? The paper (L256) mentions that PTv3 is re-benchmarked in this paper under a 'unified protocol', but claims that this increases the performance rather than decreasing it. Due to the differences between the baselines' reported performance in Tab. 1 and in the original papers, it is not clear if Point-MoE obtains such impressive results because it is truly so effective, or if it merely obtains these results because the baselines are weaker than they should be. This limits the value of the paper.

2. While Point-MoE performs quite well on average across all datasets, training it on more data does not improve performance across the board, on each individual dataset. For instance, in Tab. 1, the Structured3D performance for single-dataset training is much better than for multi-dataset training. Similarly, training on outdoor data in addition to indoor data (Tab. 1b) does not greatly impact the performance on indoor datasets (compared to Tab. 1a), and even causes a performance drop for S3DIS. These results suggest that Point-MoE still cannot fully leverage the power of massive heterogeneous datasets, which is a limitation of the approach.

3. It is not clearly explained why, in Tab. 1, the number of activated model parameters is so low for Point-MoE, compared the number of parameters of PTv3. If I understand the paper correctly, Point-MoE only replaces the projection layer after the self-attention operation with a router and experts, where each expert consists of an MLP (L075). Since two of these expert MLPs will actually be used (L347-L354), this means that the parameters from PTv3's original projection layer are replaced with the parameters from the router and the 2 MLPs, when considering only activated parameters. Since the original projection layer is quite lightweight, I assume that the router and 2 MLPs do not have fewer parameters than the projection layer. Beyond this change, all other model parameters remain the same for Point-MoE compared to PTv3. In this case, how is it possible that the total number of activated parameters for Point-MoE is so much lower than for PTv3? This should be explained better. The same applies to the drops in VRAM and FLOPs (L299).

4. There are some inconsistencies in the mathematical notation, and some symbols are not explained.
    * The symbol $N$ is used to represent three different things: the number of point clouds (L148), the number of experts (L168), and the number of points (L180). This is very confusing. The paper would be clearer if a different symbol was used to represent each of these three things.
    * Similarly, $f$ represents both the overall model in L153 and the expert function in L168.
    * In Sec. 3.2, it is not described what $\mathbf{Y}$ (Eq. 7) represents, and what it is used for.

[a] Wu et al., Point Transformer V3: Simpler, Faster, Stronger, CVPR 2024.

[b] Wu et al., Towards Large-scale 3D Representation Learning with Multi-dataset Point Prompt Training, CVPR 2024.

**Questions:**

This paper presents an idea that is original in the context of 3D segmentation, and presents a method that is effective and has clear benefits over existing approaches, but I mainly have some concerns about the low performance of the baselines. In the rebuttal, I would like to ask the authors to provide a compelling reason why the performance of the baselines is lower than what is originally reported. Additionally, I would like to ask the authors to explain why the number of activated model parameters is so low for Point-MoE, and to fix the issues with the mathematical notation. If the authors resolve these concerns, I am willing to upgrade my rating.

In addition to this, I have the following question:
* Could Point-MoE also improve the performance for single-dataset training, or it only effective for multi-dataset training? I could imagine that adaptively using specialized model parameters for different data subsets would also be useful within a single dataset. While it is beyond the main scope of the paper, I think the paper would be even stronger if it it could give some more insights into the situations in which Point-MoE is effective or not.

---

> ### Author Response · Authors · 2025-11-22
>
> ## [Q1: Performance of baselines lower than originally reported]
>
> Thanks for the great question! We attribute the performance gaps primarily to different evaluation protocols and training setups. Our goal is to obtain robust per-dataset performance from a single set of model weights, rather than maintaining N separately fine-tuned models for N datasets (which is impractical to scale in the long run). Thus, we evaluate on every dataset using the same (final) checkpoint, with no dataset-specific sampling or fine-tuning.
>
> The original PTv3 [a] is evaluated under a **closed-vocabulary** setting, which is not applicable to our setting that targets open-domain generalization to novel categories (similar in spirit to PPT). Therefore, we equip PTv3 with a CLIP head in our baseline so that it can perform zero-shot inference on unseen datasets. Furthermore, PPT [b] is proposed as a pre-training strategy and its reported numbers are obtained after dataset-specific fine-tuning; in our setup we do not use dataset labels, tailored sampling, or dataset-specific fine-tuning. For an apples-to-apples comparison, we therefore train PTv3+CLIP and PPT under the same cross-dataset protocol without dataset-specific fine-tuning. We have also tried longer training schedules and different hyperparameters for these baselines and report these efforts in Table 12 of the supplementary. This makes our protocol stricter and, we argue, more realistic, as it prioritizes cross-dataset generalization over per-dataset optimization.
>
> Our evaluation targets realistic deployment where test scenes may contain novel classes. The protocol enables: (1) zero-shot evaluation on unseen datasets (Table 2), (2) handling arbitrary class queries, and (3) a single checkpoint across all datasets without dataset labels. Since all baselines and our Point-MoE are evaluated under this same protocol, we believe the comparison is fair and reasonable.
>
> [a] Wu et al., Point Transformer V3: Simpler, Faster, Stronger, CVPR 2024.\
> [b] Wu et al., Towards Large-scale 3D Representation Learning with Multi-dataset Point Prompt Training, CVPR 2024.
>
> ## [Q2: Number of activated model parameters]
>
> We thank the reviewer for the careful reading. The key clarification is that Point-MoE-L is not built on the PTv3-L backbone. It uses the PTv3-S–style backbone (fewer layers and smaller hidden widths), whereas PTv3-L is much larger. Concretely, PTv3-L contains 42 Transformer blocks with widths up to 768, while Point-MoE-L contains 20 blocks with widths up to 512. This already halves the number of active parameters (line 258-261 in the submission). On top of this smaller backbone, Point-MoE-L adds an expert pool, but only top-2 experts are activated for each token; thus, although the model stores ≈100M total parameters, only ≈59M are activated per forward pass. PTv3-L, being fully dense, activates all ≈97M parameters per token. This explains the FLOPs (265.7 vs 375.8 GFLOPs) and VRAM (33.3 vs 37.4 GiB) reductions. Point-MoE achieves higher model capacity (100M total enabling richer specialization) with lower per-sample cost (59M activated) than dense models. We have made it more clear in the paper in Sec. 4.1 and will add a detailed parameter breakdown table in the supplementary material to improve clarity.

---

> ### Author Response · Authors · 2025-11-22
>
> ## [Q3: Single-dataset training]
>
> In our preliminary experiments (given limited rebuttal timeframe), Point-MoE on single datasets was about the same as the baseline accuracy,with a few cases of improvements such as ScanNet (75.63 v.s. 75.01) and nuScenes (78.24 v.s. 77.16) but not on Structured3D (79.29 v.s. 79.37), Matterport3D (53.47 v.s. 53.89) and Waymo (69.50 v.s. 70.20). We believe we may need to introduce load balancing loss since individual datasets in the point cloud domain do not have significant diversity (same sensor configuration, same location, similar distribution of objects), and therefore may not organically form meaningful data subsets. Therefore we hypothesize that the additional expressive capacity of the MoE paradigm remains underutilized within single datasets, whereas we see consistent improvements in the multi-dataset setting, which has larger scale and more data diversity. Definitely a direction worth looking into, with more rigorous empirical studies, before we can make a strong conclusion. We will include this discussion in the appendix, and we agree with the reviewer that it is important for readers and practitioners in the field to have clear insights into where Point-MoE is best applicable, and where it is not.
>
> | Method    | ScanNet   | Structured3D | Matterport3D | nuScenes  | Waymo     | Average   |
> | --------- | --------- | ------------ | ------------ | --------- | --------- | --------- |
> | PTv3      | 75.01     | **79.37**    | **53.89**    | 77.16     | **70.20** | 71.13     |
> | Point-MoE | **75.63** | 79.29        | 53.47        | **78.24** | 69.50     | **71.23** |
>
> ## [W2: Training on more data does not improve performance across the board]
>
> Thanks for pointing it out! Indeed, almost all multi-dataset methods for point clouds have had to contend with this challenge. Although we have achieved overall best performance across multiple datasets, Point-MoE and other baseline methods (e.g. PPT)  lag behind dataset-specific fine-tuned models. However, Point-MoE gives us the advantage of having a *single* set of model weights working across multiple datasets, instead of having to maintain a set of model weights for each individual dataset, which naturally is not a scalable and generalizable solution as we scale up the data sources.
>
> Moreover, we identify that the CLIP head (a design choice to support open-vocab generalization to unseen class ontologies in the zero-shot setting; also used in PPT) as one of the bottlenecks that hinders performance on Structured3D. Certain classes in Structured3D are hard to distinguish in the CLIP semantic space, such as *otherstructure*, *otherfurniture* and *otherprop*. To test out this hypothesis, we freeze the backbone of a jointly-pretrained PointMoE-L and further the CLIP head with a linear classifier. We find modest increases for ScanNet (76.0 v.s. 77.1) and S3DIS (69.5 v.s. 70.4) compared to the original Point-MoE, but Structured3D benefits from this with a **7.8 mIoU** increase. Although this still lags behind a single-dataset model’s performance (81.5 v.s. 75.5), the gap is closed significantly. We note that all of our models and baselines are evaluated using CLIP head in a consistent apples-to-apples manner, so relative deltas and trends are still meaningful.
>
> ## [W4: Inconsistencies in the mathematical notation]
>
> Thanks for pointing it out, and we apologize for the miss! We have updated the manuscript and explained them in detail accordingly.

---

> ### Comment · Reviewer_wdW6 · 2025-11-25
>
> Thanks for the extensive response to my questions and concerns. With this response, several of my concerns have been addressed. However, I have some remaining questions:
>
> * **Regarding the performance of PPT**: Thanks for clarifying that the original PPT paper reports performance after dataset-specific finetuning, while this paper reports performance after joint training only. However, when further inspecting the original PPT paper [a], I found that they report only a minor difference between the performance of PPT after (a) supervised joint training or (b) supervised pre-training + dataset-specific fine-tuning, in combination with the SparseUNet model.
>     * Is the reported "PPT Sup. (joint)" setting in the PPT paper the same setting that is used for supervised joint training in the submitted paper, besides the use of SparseUNet instead of PTv3?
>     * If so, how do the authors explain that there is only a small performance difference between "PPT Sup. (joint)" and "PPT Sup. (f.t.)" in the original paper, while the difference between the reported PPT performance in this paper (supervised joint training) and the original PPT performance (using the "PPT Sup. (f.t.)" setting) is so large, e.g., 10.0 mIoU difference on S3DIS?
>
> [a] Wu et al., Towards Large-scale 3D Representation Learning with Multi-dataset Point Prompt Training, CVPR 2024.
>
> * **Regarding the performance of PTv3**: Thanks for clarifying that a difference between PTv3 and Point-MoE is that Point-MoE uses a CLIP classification head, similar to PPT, instead of a linear head. However, the PPT paper [a] shows in Tab. 13c that this CLIP-based "language guided categorical alignment" head actually outperforms a linear head in a single-dataset segmentation setting. This finding conflicts with the claim in the author response that replacing the linear head with the CLIP head causes a lower performance for single-dataset segmentation with PTv3.
>      * In light of these findings in the PPT paper, do the authors still believe the difference in performance of PTv3 in this submitted manuscript and the original PTv3 paper is due to the use of a CLIP-based segmentation head? If so, how do they explain the conflicting results?
>
> [a] Wu et al., Towards Large-scale 3D Representation Learning with Multi-dataset Point Prompt Training, CVPR 2024.
>
> * **Regarding the number of parameters**: Thanks for clarifying that Point-MoE-L is not based on PTv3-L and therefore does not have the same number of layers, I misunderstood this. However, there seem to be some minor conflicting statements between the author response and the paper. The author response mentions that PTv3-L contains 42 Transformer blocks while L243 of the paper states that it uses 18+12=30 layers. Similarly, the author response mentions 20 blocks for Point-MoE-L but L244 of the paper states 14+8=22 layers.
>     * Which of these is correct?
>
> Finally, thank you for improving the mathematical notation. However, I noticed that my comment about the meaning of $\mathbf{Y}$ from Eq. 7 is not yet addressed.
>
> I look forward to reading your answers to my questions. Thanks in advance.
>
> Best,
> Reviewer wdW6

---

> ### Author Response · Authors · 2025-11-28
>
> ## [Q1: Regarding performance of PPT]
> Besides the difference in backbones, there are several other critical differences that further explains the gap between our PPT results and those reported in the original paper:
> - (1) **Oracle domain labels vs. domain detector.** The original PPT assumes access to ground-truth domain labels during both training and inference, enabling the model to adapt features using perfectly accurate dataset identities. In practice, such labels are unavailable at test time. Therefore, we introduce a domain detector (#233-237, Supp. B.2), inspired by One-for-All. Although accurate, it misclassifies S3DIS as ScanNet ~6% of the time. When we restore oracle domain labels at test time, PPT-L improves from 65.0 → 72.3 mIoU on S3DIS which is already close to the original PPT number (72.2 mIoU under the "Sup. joint" setting), showing that the discrepancy is primarily due to this domain detector.
> - (2) **Dataset-specific vs. uniform sampling.** The original PPT employs a hand-tuned 4:2:1 sampling ratio between the datasets in the training set (Tab. 2), which boosts performance by ~1 mIoU over uniform sampling over datasets. For fair comparison across PTv3, PPT, and Point-MoE, and for scalability to many datasets, we use uniform sampling, avoiding dataset-specific tuning and removing a confounding factor.
>
> Finally, we want to emphasize that training a domain detector for truly in-the-wild, large-scale scenarios is not practical. Specifically, an unseen data sample may not correctly map to any seen dataset. Point-MoE offers a way to train and perform inference without an explicit notion of domain, by dynamically routing tokens in a self-organizing manner (Fig. 2). We hope this clarifies the misunderstanding.
>
> ## [Q2: Regarding the performance of PTv3]
> We thank the reviewer for highlighting the LCA ablation in the PPT paper (Tab. 13c). Using the official code for both experiments, we re-ran training on S3DIS for 3,000 epochs in two settings: (i) the linear head and (ii) the LCA (CLIP) head, where we only swap the linear head with LCA head while keeping all other components unchanged. The linear head reaches 71.9 mIoU, and the LCA head achieves 67.6 mIoU. Our 71.9 is lower than the PPT report; due to the limited rebuttal window we used a larger batch size, which likely contributes to this gap. Importantly, these results indicate that LCA does not consistently yield improvements across backbones. We believe this complements the LCA ablation, since the PPT paper’s ablation evaluates one backbone (SparseUNet) and one dataset, whereas our finding suggests that replacing a linear head with LCA may not directly transfer gains. We hypothesize LCA benefited SparseUNet (small) as demonstrated in PPT paper (Tab. 13c) partly due to capacity, since an LCA head has ~50× more parameters than a linear head. In our experiments with PTv3 as backbone, LCA head might be more sensitive to hyperparameters, since we find it plateaus even with more training epochs.
>
> ## [Q3: Regarding the number of parameters]
>
> Apology for the confusion! The version in our paper is actually correct, and we had a typo in the last response. PTv3-L contains 30 layers and Point-MoE-L contains 22 layers.
>
> ## [Notation]
>
> Thanks for catching this! We have further revised the manuscript.

---

### Official Review · Reviewer_tPuN · 2025-10-31

**Soundness:** 3
**Presentation:** 3
**Contribution:** 3
**Rating:** 6
**Confidence:** 4

**Summary:**

The paper explores the shift from dense to sparse Transformers for point cloud processing and investigates mixed indoor-outdoor training with a MoE framework. The approach aims to leverage heterogeneous data for improved generalization and domain adaptation. The results demonstrate notable accuracy gains across datasets and suggest that the MoE design effectively mitigates cross-dataset interference.

**Strengths:**

the authors present evidence that mixed-data training enhances performance on each constituent dataset. The MoE framework appears effective at isolating domain-specific patterns while suppressing cross-dataset interference, thereby amplifying the positive effects of data scaling. This insight is valuable.

**Weaknesses:**

1. The shift from dense to sparse Transformers is a significant architectural choice; however, the paper would benefit from a more thorough analysis of its practical implications on point cloud data — particularly regarding training/inference latency overhead and training stability. These aspects are critical for assessing whether the reported accuracy gains come with hidden trade-offs (e.g., convergence difficulty, increased wall-clock time, or sensitivity to hyperparameters).
2. The experiments focus on heterogeneous indoor-outdoor training and inference. But several key questions remain underexplored:
- Do different sensor modalities (or sensor configurations) benefit equally from the proposed approach?
- Are the learned experts equally capable of capturing and adapting to sensor-specific data distributions, or do some sensors dominate or degrade performance?
3. Indoor and outdoor scenes differ substantially in object geometry, scale, density, and context. Training across both domains clearly leverages additional data — but what specific capabilities does this impart to the model? Is it improved generalization, better feature disentanglement, or simply regularization?
4. Mixed-dataset training yields a performance gap between indoor and outdoor evaluation: outdoor performance lags significantly behind indoor gains. This raises an important concern — could indoor data be introducing noise or domain bias that hampers scaling benefits for outdoor scenes, rather than helping them?
5.  The observed zero-shot gains are markedly stronger in indoor settings. Is this truly due to better scene or acquisition-mode discrimination — or might it instead reflect the MoE’s improved ability to specialize on object-level distinctions that happen to be more prevalent or cleanly separable indoors?

**Questions:**

See weaknesses.

---

> ### Author Response · Authors · 2025-11-22
>
> ## [Q1:  Training/inference latency overhead and training stability]
>
> We thank the reviewer for this question. Point-MoE demonstrates no hidden trade-offs to our knowledge: inference is actually more efficient than dense baselines.
>
> **Inference efficiency**: Per Table 11 in Supplementary materials, Point-MoE-L reduces FLOPs by 30.9% (265.7 vs 375.8 GFLOPs) and peak memory by 11-19% (33.3 vs 37.4 GiB) compared to PTv3-L (Table 11), making deployment faster despite higher accuracy.
>
> **Training overhead:** Point-MoE-L is modestly slower (1.19s vs 0.86s per step) and uses slightly more memory (54.6 vs 44 GiB) due to routing. This training cost is acceptable given +8.1 mIoU gains and 30% faster inference.
>
> **Training stability**: We use identical hyperparameters as baselines (Table 9) and observe smooth, monotonic convergence without divergence. Table 3 shows robust performance across batch sizes (3i), top-k values (3b), and expert counts (3h), indicating low hyperparameter sensitivity. Notably, removing auxiliary load-balancing loss **improves** performance (3a: 70.4 vs 69.8), demonstrating natural expert specialization without manual tuning.
>
> ## [Q2: Different sensor modalities benefit & experts capturing and adapting to sensor-specific data distributions]
>
> **(a)** To directly test whether gains are dominated by a particular sensor type, we train indoor-only and outdoor-only variants in addition to the full mixed setting, and then regroup results by sensor modality. Within the **indoor** domain, Structured3D has a different distribution (synthetic rendering) compared with the two RGB-D reconstruction datasets; all three see clear gains between +3.1% and +24.1% mIoU (see Table 1 below). Within the **outdoor** domain, SemanticKITTI and nuScenes use different LiDAR configurations (64-beam vs 32-beam), and both improve similarly (See Table 2 below). **Zero-shot** evaluation on **unseen sensor configuration**s also shows consistent gains (See Table 3 below). Thus, we observe consistent gains across sensor configurations from Point-MoE, both in-domain and out-of-domain.
>
> | Indoor Sensor Modality | Datasets     | PTv3-L | Point-MoE-L | Gain  | %Improvement |
> | ---------------------- | ------------ | ------ | ----------- | ----- | ------------ |
> | RGB-D Reconstruction   | ScanNet      | 72.2   | 76.0        | +3.8  | +5.2%        |
> | Synthetic Rendering    | Structured3D | 56.0   | 69.5        | +13.5 | +24.1%       |
> | RGB-D Reconstruction   | S3DIS        | 67.4   | 69.5        | +2.1  | +3.1%        |
>
> | Outdoor Sensor Modality | Datasets      | PTv3-L | Point-MoE-L | Gain | %Improvement |
> | ----------------------- | ------------- | ------ | ----------- | ---- | ------------ |
> | 64-beam LiDAR           | SemanticKITTI | 63.1   | 67.2        | +4.1 | +6.5%        |
> | 32-beam LiDAR           | nuScenes      | 70.4   | 74.2        | +3.8 | +5.4%        |
>
> Zero-shot generalization to unseen sensor configurations:
>
> | Sensor Modality      | Datasets     | PTv3-L | Point-MoE-L | Gain | %Improvement |
> | -------------------- | ------------ | ------ | ----------- | ---- | ------------ |
> | RGB-D Reconstruction (different device) | Matterport3D | 39.1   | 41.3        | +2.2 | +5.6%        |
> | 64-beam LiDAR  (difference range)      | Waymo        | 21.9   | 23.7        | +1.8 | +8.2%        |
>
> **(b)** We visualize expert choices in Fig. 4 and report sensor modalities in Tab. 6. For each MoE layer, we compute the Jensen–Shannon divergence (JSD) between expert-selection distributions of different datasets and observe consistently high JSD across modalities, indicating that Point-MoE uses different mixtures of experts for different sensors (e.g., LiDAR vs RGB-D) rather than letting one modality dominate. At the same time, all sensor types see non-degrading or improved mIoU compared to PTv3 baseline, so no modality is harmed by MoE. We note that fully disentangling “sensor effect” from “content effect” would require a dataset where the same scenes are captured with multiple sensors, which we view as interesting future work.

---

> ### Author Response · Authors · 2025-11-22
>
> ## [Q3: How training across both domains impact the model]
>
> We thank the reviewer for this question. Reformulating Table 2 for Matterport3D, we see that *adding outdoor data improves zero-shot performance* for Point-MoE-L from 43.9 → 45.5 mIoU (+1.6), while PTv3-L changes only slightly (42.2 → 42.5, +0.3) and PPT-L actually degrades (27.1 → 24.0, −3.1). This indicates that, for Point-MoE, mixed indoor–outdoor training goes beyond simple regularization and yields better domain generalization to unseen indoor scenes.
>
> | Training data: →Models: ↓ | Val (In) | Test (In) | Avg (In) | Val (In+out) | Test (In + out) | Avg(In + out) | Delta |
> | ------------------------- | -------- | --------- | -------- | ------------ | --------------- | ------------- | ----- |
> | PTv3-L                    | 39.0     | 45.4      | 42.2     | 39.1         | 45.9            | 42.5          | +0.3  |
> | PPT-L                     | 27.8     | 26.3      | 27.1     | 24.9         | 23.1            | 24.0          | -3.1  |
> | Point-MoE-L               | 41.0     | 46.7      | 43.9     | 41.3         | 49.6            | 45.5          | +1.6  |
>
> To probe disentanglement, we measure the mean Jensen–Shannon divergence (JSD) between expert-selection distributions for different datasets. JSD is low when all datasets use similar experts (minimal specialization) and increases when different datasets consistently route to different experts (more specialization), which can indicate disentanglement in the feature space. With indoor-only training, the mean decoder JSD is 0.10, whereas with indoor+outdoor training it rises to 0.34, indicating that outdoor data encourages experts to specialize more strongly across domains and sensor types.
>
> Together, the improved zero-shot mIoU and higher inter-dataset JSD provide evidence that mixed-domain training improves both *generalization* and *feature disentanglement*, beyond simply acting as *regularization*.
>
>
> ## [Q4: Indoor data introduces noise or domain bias]
>
> Referring to Table 1(b), in the indoor+outdoor joint setting Point-MoE-L consistently outperforms PTv3-L by 1.8, 5.4, 6.3, 0.1, 3.3, and 3.5 mIoU on ScanNet, Structured3D, S3DIS, nuScenes, and SemanticKITTI, respectively. Averaged across datasets, the gain on indoor datasets is +3.4 mIoU, and the gain on outdoor datasets is +3.35 mIoU which is essentially identical to indoor gain. This shows that mixed-dataset training helps both domains to a similar degree, and indoor data does not introduce significant noise or bias that suppresses overall outdoor performance.
>
> Note that PPT-L additionally uses dataset-prompt-driven normalization, so the improvements of Point-MoE-L over PTv3-L isolate the contribution of the MoE design rather than relying on domain-specific cues.
>
> ## [Q5: Zero-shot gains stronger in indoor settings]
>
> We appreciate this question. The zero-shot gains appear larger for novel indoor scenes mainly because the outdoor zero-shot benchmark (Waymo) contains many semantic categories that never appear in the training outdoor datasets (SemanticKITTI, nuScenes). Specifically, Waymo evaluates 22 classes, of which 9 are completely unseen during training (Table 8). When we compute zero-shot performance only on the 13 shared categories, Point-MoE-L achieves ≈40.1% mIoU, which is comparable to our zero-shot indoor Matterport3D results.
>
> Thus, we believe, the difference in reported numbers is largely due to taxonomy mismatch, not because MoE specializes better in indoor datasets.

---

### Author Response · Authors · 2025-12-03
**Summary for AC**

**Key Contributions**: Point-MoE enables multi-dataset 3D semantic segmentation across diverse indoor–outdoor domains by routing tokens through a Mixture-of-Experts, rather than relying on normalization-based multi-dataset approaches. It achieves the best average mIoU on in-domain datasets (Table 1) and generalizes robustly to unseen domains (Table 2), with routing behavior analyzed in Fig. 2 and Fig. 3. Unlike PPT [a], which requires dataset/domain labels during training and a domain detector at inference, Point-MoE attains stronger performance (+3.9 mIoU compared to PPT) with ~30% fewer FLOPs, without access to domain labels. Importantly, Point-MoE uses a single set of model weights that maintains strong performance across multiple seen datasets and generalizes to unseen datasets, without any per-dataset fine-tuning, avoiding the need to maintain N separate models for N datasets.

**Reviewers have these common positive thoughts about our submission**:
- (1) A clear and practically useful formulation of multi-dataset 3D semantic segmentation without dataset labels at inference (R3, R4).
- (2) An original, well-motivated MoE-based architecture that dynamically routes tokens to experts (R2, R3, R4).
- (3) Strong empirical performance (R1, R2, R3, R4).
- (4) Thorough and insightful ablations and analyses of MoE design choices and expert specialization (R2, R3, R4).

Here we summarize our interactions with each review in order to aid the decision of AC:

**R1 (rating 6):**

**Summarized Questions**: This reviewer asked about practical trade-offs of moving from dense to sparse PTv3, whether gains are consistent across sensor modalities, and what mixed indoor–outdoor training truly buys (generalization vs. regularization).

**Summarized Answer**: We provide additional efficiency analysis and we show consistent gains across all indoor and outdoor modalities, as well as zero-shot sensors. Lastly, We demonstrate Point-MoE improves zero-shot generalization and increases expert specialization via JSD analysis.

**R2 (rating 6)**: This reviewer stated *"willing to upgrade if concerns are resolved"* and asked 2 follow-up questions as we resolved other concerns.

**Summarized Follow-up Questions**: The reviewer asked additional questions about the PPT performance discrepancy with the original paper, the role of the CLIP/LCA head in our PTv3 results.

**Summarized Answer**: We carefully addressed both follow-up concerns. The first concern is a misunderstanding and we have run additional experiments to confirm the gap of CLIP head is reproducible.

**R3 (rating 6)**:

**Summarized Questions**: This reviewer asked how routing works at inference, how CLIP-based alignment interacts with MoE experts, why we remove the load-balancing loss, and whether our gains are mainly engineering given model complexity.

**Summarized Answers**: We clarified misunderstandings about MoE related questions. We showed that dropping load balancing is empirically beneficial and supported by recent MoE work. Lastly, we argued that Point-MoE provides significant gains from architectural design choices, beyond just engineering over a strong PTv3 baseline and PPT, including large zero-shot improvements.

**R4 (rating 4)**:

**Summarized Follow-up Questions**: The reviewer has additional concerns about (1) missing comparisons and (2) not being single dataset SOTAs.

**Summarized Answers**: We clarified that Point-MoE is trained under multi-dataset 3D segmentation setting with a single set of model weight across indoor–outdoor point clouds and no dataset labels at train or test time, which is complementary to existing open-vocabulary and multi-dataset methods. Unlike most contemporary approaches, we explicitly aim to use a single set of model weights that (a) maintains strong performance across multiple seen datasets (mitigating negative transfer in multi-domain training) and (b) generalizes robustly to unseen datasets, instead of maintaining N separate models fine-tuned for N datasets, which is unscalable in the long run. We respectfully disagreed with both concerns, emphasizing that we benchmark against strong, relevant baselines (PTv3, PPT) and show consistent in-domain and zero-shot gains under this single-backbone, domain-label-free setting.

[a]. Towards Large-scale 3D Representation Learning with Multi-dataset Point Prompt Training

---

### Meta-Review · Area_Chair_HsPE · 2026-01-06

**Summary:**

This paper proposes PointMoE (mixture-of-experts) for large-scale multi-dataset 3D semantic segmentation. The method uses routing to enable dynamic expert specialization across datasets. The approach does not need to specify which dataset is used, and a single neural network can perform zero-shot semantic label prediction to an arbitrary type of point cloud.

**Reviewer Concerns:**

The reviewers generally appreciated that the motivation is practical since it can handle heterogeneity without domain labels. Major concerns are summarized below.

1. Efficiency of Parameters (wdW6, 9WmA): The reviewers asked for clarification of activated parameters and the inference-time latency.
2. Design choice (rFAr): The reviewer questioned why load-balancing loss was removed
3. Not a top-performing performance for single-dataset (9WmA, wdW6): The reviewers mentioned that the reported performance is lagging behind recent approaches, such as UniDet3D, ArKit LabelMaker, and Sonata. In addition, The reviewer asks why PTv3 and PPT performance were lower than the reported number in the original paper.
4. Evaluation protocol (9WmA): The reviewer mentioned that approaches that tackle heterogeneous 3D datasets exist, and the paper's main claim and evaluation are misleading.

**Reviewer Scores:**

The paper received an initial score of 6, 6, 6, and 4 from four reviewers. To address each reviewer's concerns, the authors provide the following extensive feedback.

For handling 1, the authors clarified that the backbone has 30% fewer FLOPs (compared to PTv3-L) and lower active parameters during inference despite a higher total parameter count. For 2, the authors mention a recent paper that explains that dataset imbalance degrades performance. Regarding 3, the authors explained that, under the unified, stricter configuration, they used a single checkpoint across all datasets, which brings a trade-off. Regarding 4, the authors explain that the approaches mentioned have different setups (e.g., CUA-O3D is based on distilling 2D models), whereas the proposed approach accounts for point-level heterogeneity.

Overall, AC notes that the authors provide an extensive rebuttal, and AC confirms that most of the reviewers' concerns can be addressed appropriately. Therefore, AC recommends the acceptance of the paper with strong confidence. AC requests that authors apply the questions and discussions to the revised version. For example, it is necessary to carefully highlight how the reported performance of other approaches, which is inconsistent in their original paper, can be interpreted and how the evaluation setup and protocol differ from those of other papers.

---

### Decision · Program_Chairs · 2026-01-26

Accept (Poster)